# A CRITICAL ANALYSIS OF DISTRIBUTION SHIFT

## ABSTRACT

We introduce three new robustness benchmarks consisting of naturally occurring distribution changes in image style, geographic location, camera operation, and more. Using our benchmarks, we take stock of previously proposed hypotheses for out-of-distribution robustness and put them to the test. We find that using larger models and synthetic data augmentation can improve robustness on real-world distribution shifts, contrary to claims in prior work. Motivated by this, we introduce a new data augmentation method which advances the state-of-the-art and outperforms models pretrained with $1000\times$ more labeled data. We find that synthetic augmentations can sometimes improve real-world robustness. We also find that some methods consistently help with distribution shifts in texture and local image statistics, but these methods do not help with some other distribution shifts like geographic changes. Hence no evaluated method consistently improves robustness. We conclude that future research must study multiple distribution shifts simultaneously.

## 1 INTRODUCTION

While the research community must create robust models that generalize to new scenarios, the robustness literature (Dodge and Karam, 2017; Geirhos et al., 2020) lacks consensus on evaluation benchmarks and contains many dissonant hypotheses. Hendrycks et al. (2020a) find that many recent language models are already robust to many forms of distribution shift, while Yin et al. (2019) and Geirhos et al. (2019) find that vision models are largely fragile and argue that data augmentation offers one solution. In contrast, Taori et al. (2020) provide results suggesting that using pretraining and improving in-distribution test set accuracy improve natural robustness, whereas other methods do not.

In this paper we articulate and systematically study seven robustness hypotheses. The first four hypotheses concern *methods* for improving robustness, while the last three hypotheses concern abstract *properties* about robustness. These hypotheses are as follows.

- *Larger Models*: increasing model size improves robustness (Hendrycks and Dietterich, 2019; Xie and Yuille, 2020).
- *Self-Attention*: adding self-attention layers to models improves robustness (Hendrycks et al., 2019b).
- *Diverse Data Augmentation*: robustness can increase through data augmentation (Yin et al., 2019).
- *Pretraining*: pretraining on larger and more diverse datasets improves robustness (Orhan, 2019; Hendrycks et al., 2019a).
- *Texture Bias*: convolutional networks are biased towards texture, which harms robustness (Geirhos et al., 2019).
- *Only IID Accuracy Matters*: accuracy on independent and identically distributed test data entirely determines natural robustness.
- *Synthetic $\not\Rightarrow$ Real*: *synthetic* robustness interventions including diverse data augmentations do not help with robustness on *real-world* distribution shifts (Taori et al., 2020).

It has been difficult to arbitrate these hypotheses because existing robustness datasets preclude the possibility of controlled experiments by varying multiple aspects simultaneously. For instance, *Texture Bias* was initially investigated with synthetic distortions (Geirhos et al., 2018), which conflicts with the *Synthetic $\not\Rightarrow$ Real* hypothesis. On the other hand, natural distribution shifts often affect many factors (e.g., time, camera, location, etc.) simultaneously in unknown ways (Recht et al., 2019;

Figure 1: Images from our three new datasets ImageNet-Renditions (ImageNet-R), DeepFashion Remixed (DFR), and StreetView StoreFronts (SVSF). The SVSF images are recreated from the public Google StreetView, copyright Google 2020. Our datasets test robustness to various naturally occurring distribution shifts including rendition style, camera viewpoint, and geography.

Hendrycks et al., 2019b). Existing datasets also lack diversity such that it is hard to extrapolate which methods will improve robustness more broadly. To address these issues and test the seven hypotheses outlined above, we introduce three new robustness benchmarks and a new data augmentation method.

First we introduce ImageNet-Renditions (ImageNet-R), a 30,000 image test set containing various renditions (e.g., paintings, embroidery, etc.) of ImageNet object classes. These renditions are naturally occurring, with textures and local image statistics unlike those of ImageNet images, allowing us to more cleanly separate the *Texture Bias* and *Synthetic $\Longrightarrow$ Real* hypotheses.

Next, we investigate natural shifts in the image capture process with StreetView StoreFronts (SVSF) and DeepFashion Remixed (DFR). SVSF contains business storefront images taken from Google Streetview, along with metadata allowing us to vary location, year, and even the camera type. DFR leverages the metadata from DeepFashion2 (Ge et al., 2019) to systematically shift object occlusion, orientation, zoom, and scale at test time. Both SVSF and DFR provide distribution shift controls and do not alter texture, which remove possible confounding variables affecting prior benchmarks.

Finally, we contribute DeepAugment to increase robustness to some new types of distribution shift. This augmentation technique uses image-to-image neural networks for data augmentation, not data-independent Euclidean augmentations like image shearing or rotating as in previous work. DeepAugment achieves state-of-the-art robustness on our newly introduced ImageNet-R benchmark and a corruption robustness benchmark. DeepAugment can also be combined with other augmentation methods to outperform a model pretrained on $1000\times$ more labeled data.

After examining our results on these three datasets and others, we can rule out several of the above hypotheses while strengthening support for others. As one example, we find that synthetic data augmentation robustness interventions improve accuracy on ImageNet-R and real-world image blur distribution shifts, providing clear counterexamples to *Synthetic $\Longrightarrow$ Real* while lending support to the *Diverse Data Augmentation* and *Texture Bias* hypotheses. In the conclusion, we summarize the various strands of evidence for and against each hypothesis. Across our many experiments, we do not find a general method that consistently improves robustness, and some hypotheses require additional qualifications. While robustness is often spoken of and measured as a single scalar property like accuracy, our investigations suggest that robustness is not so simple. In light of our results, we hypothesize in the conclusion that robustness is *multivariate*.

## 2 RELATED WORK

**Robustness Benchmarks.** Recent works (Hendrycks and Dietterich, 2019; Recht et al., 2019; Hendrycks et al., 2020a) have begun to characterize model performance on out-of-distribution (OOD) data with various new test sets, with dissonant findings. For instance, Hendrycks et al. (2020a) demonstrate that modern language processing models are moderately robust to numerous naturally occurring distribution shifts, and that *Only IID Accuracy Matters* is inaccurate for natural language

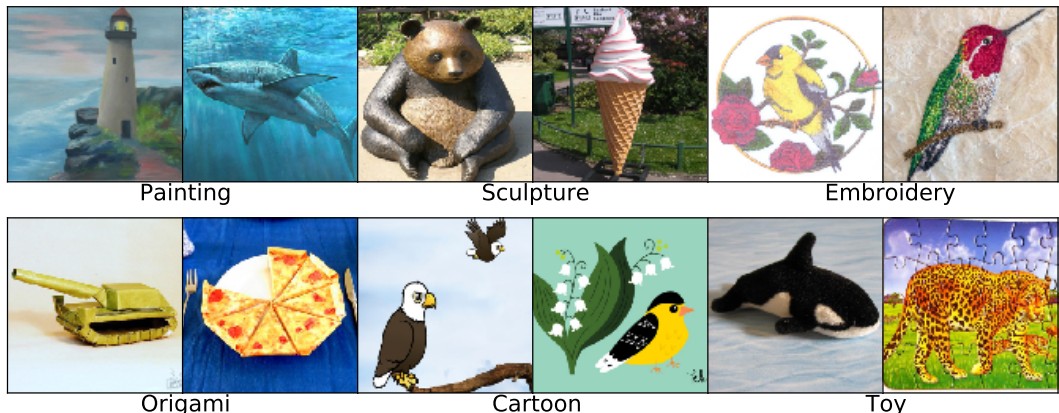

Figure 2: ImageNet-Renditions (ImageNet-R) contains 30,000 images of ImageNet objects with different textures and styles. This figure shows only a portion of ImageNet-R's numerous rendition styles. The rendition styles (e.g., "Toy") are for clarity and are *not* ImageNet-R's classes; ImageNet-R's classes are a subset of 200 ImageNet classes. ImageNet-R emphasizes shape over texture.

tasks. For image recognition, Hendrycks and Dietterich (2019) analyze image models and show that they are sensitive to various simulated image corruptions (e.g., noise, blur, weather, JPEG compression, etc.) from their "ImageNet-C" benchmark.

Recht et al. (2019) reproduce the ImageNet (Russakovsky et al., 2015) validation set for use as a benchmark of naturally occurring distribution shift in computer vision. Their evaluations show a 11-14% drop in accuracy from ImageNet to the new validation set, named ImageNetV2, across a wide range of architectures. Taori et al. (2020) use ImageNetV2 to measure natural robustness and dismiss *Diverse Data Augmentation*. Recently, Engstrom et al. (2020) identify statistical biases in ImageNetV2's construction, and they estimate that reweighting ImageNetV2 to correct for these biases results in a less substantial 3.6% drop.

**Data Augmentation.** Geirhos et al. (2019); Yin et al. (2019); Hendrycks et al. (2020b) demonstrate that data augmentation can improve robustness on ImageNet-C. The space of augmentations that help robustness includes various types of noise (Madry et al., 2017; Rusak et al., 2020; Lopes et al., 2019), highly unnatural image transformations (Geirhos et al., 2019; Yun et al., 2019; Zhang et al., 2017), or compositions of simple image transformations such as Python Imaging Library operations (Cubuk et al., 2018; Hendrycks et al., 2020b). Some of these augmentations can improve accuracy on in-distribution examples as well as on out-of-distribution (OOD) examples.

## 3 NEW BENCHMARKS

In order to evaluate the seven robustness hypotheses, we introduce three new benchmarks that capture new types of naturally occurring distribution shifts. ImageNet-Renditions (ImageNet-R) is a newly collected test set intended for ImageNet classifiers, whereas StreetView StoreFronts (SVSF) and DeepFashion Remixed (DFR) each contain their own training sets and multiple test sets. SVSF and DFR split data into a training and test sets based on various image attributes stored in the metadata. For example, we can select a test set with images produced by a camera different from the training set camera. We now describe the structure and collection of each dataset.

### 3.1 IMAGENET-RENDITIONS (IMAGENET-R)

While current classifiers can learn some aspects of an object's shape (Mordvintsev et al., 2015), they nonetheless rely heavily on natural textural cues (Geirhos et al., 2019). In contrast, human vision can process abstract visual renditions. For example, humans can recognize visual scenes from line drawings as quickly and accurately as they can from photographs (Biederman and Ju, 1988). Even some primates species have demonstrated the ability to recognize shape through line drawings (Itakura, 1994; Tanaka, 2006).

To measure generalization to various abstract visual renditions, we create the ImageNet-Rendition (ImageNet-R) dataset. ImageNet-R contains various artistic renditions of object classes from the original ImageNet dataset. Note the original ImageNet dataset discouraged such images since annotators were instructed to collect "photos only, no painting, no drawings, etc." (Deng, 2012). We do the opposite.

**Data Collection.** ImageNet-R contains 30,000 image renditions for 200 ImageNet classes. We choose a subset of the ImageNet-1K classes, following Hendrycks et al. (2019b), for several reasons. A handful ImageNet classes already have many renditions, such as "triceratops." We also choose a subset so that model misclassifications are egregious and to reduce label noise. The 200 class subset was also chosen based on rendition prevalence, as "strawberry" renditions were easier to obtain than "radiator" renditions. Were we to use all 1,000 ImageNet classes, annotators would be pressed to distinguish between Norwich terrier renditions as Norfolk terrier renditions, which is difficult. We collect images primarily from Flickr and use queries such as "art," "cartoon," "graffiti," "embroidery," "graphics," "origami," "painting," "pattern," "plastic object," "plush object," "sculpture," "line drawing," "tattoo," "toy," "video game," and so on. Images are filtered by Amazon MTurk annotators using a modified collection interface from ImageNetV2 (Recht et al., 2019). For instance, after scraping Flickr images with the query "lighthouse cartoon," we have MTurk annotators select true positive lighthouse renditions. Finally, as a second round of quality control, graduate students manually filter the resulting images and ensure that individual images have correct labels and do not contain multiple labels. Examples are depicted in Figure 2. ImageNet-R also includes the line drawings from Wang et al. (2019), excluding horizontally mirrored duplicate images, pitch black images, and images from the incorrectly collected "pirate ship" class.

## 3.2 STREETVIEW STOREFRONTS (SVSF)

Computer vision applications often rely on data from complex pipelines that span different hardware, times, and geographies. Ambient variations in this pipeline may result in unexpected performance degradation, such as degradations experienced by health care providers in Thailand deploying laboratory-tuned diabetic retinopathy classifiers in the field (Beede et al., 2020). In order to study the effects of shifts in the image capture process we collect the StreetView StoreFronts (SVSF) dataset, a new image classification dataset sampled from Google StreetView imagery (Anguelov et al., 2010) focusing on three distribution shift sources: country, year, and camera.

**Data Collection.** SVSF consists of cropped images of business store fronts extracted from StreetView images by an object detection model. Each store front image is assigned the class label of the associated Google Maps business listing through a combination of machine learning models and human annotators. We combine several visually similar business types (e.g. drugstores and pharmacies) for a total of 20 classes, listed Appendix B.

Splitting the data along the three metadata attributes of country, year, and camera, we create one training set and five test sets. We sample a training set and an in-distribution test set (200K and 10K images, respectively) from images taken in US/Mexico/Canada during 2019 using a "new" camera system. We then sample four OOD test sets (10K images each) which alter one attribute at a time while keeping the other two attributes consistent with the training distribution. Our test sets are year: 2017, 2018; country: France; and camera: "old."

## 3.3 DEEPFASHION REMIXED

Changes in day-to-day camera operation can cause shifts in attributes such as object size, object occlusion, camera viewpoint, and camera zoom. To measure this, we repurpose DeepFashion2 (Ge et al., 2019) to create the DeepFashion Remixed (DFR) dataset. We designate a training set with 48K images and create eight out-of-distribution test sets to measure performance under shifts in object size, object occlusion, camera viewpoint, and camera zoom-in. DeepFashion Remixed is a multi-label classification task since images may contain more than one clothing item per image.

**Data Collection.** Similar to SVSF, we fix one value for each of the four metadata attributes in the training distribution. Specifically, the DFR training set contains images with medium scale, medium occlusion, side/back viewpoint, and no zoom-in. After sampling an IID test set, we construct eight

Original        DeepAugment

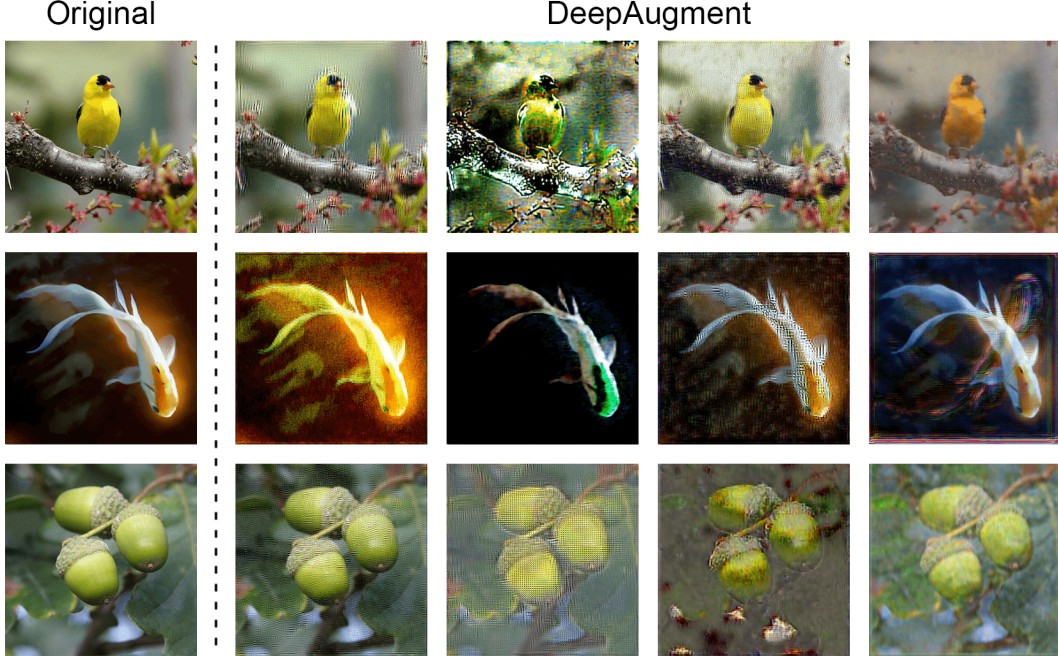

Figure 3: DeepAugment examples preserve semantics, are data-dependent, and are far more visually diverse than augmentations such as rotations.

OOD test distributions by altering one attribute at a time, obtaining test sets with minimal and heavy occlusion; small and large scale; frontal and not-worn viewpoints; and medium and large zoom-in. See Appendix B for details on test set sizes.

## 4 DEEPAUGMENT

In order to further explore the *Diverse Data Augmentation* hypothesis, we introduce a new data augmentation technique. Whereas most previous data augmentations techniques use simple augmentation primitives applied to the raw image itself, we introduce DeepAugment, which distorts images by perturbing internal representations of deep networks.

DeepAugment works by passing a clean image through an image-to-image network and introducing several perturbations during the forward pass. These perturbations are randomly sampled from a set of manually designed functions and applied to the network weights and to the feed-forward signal at random layers. For example, our set of perturbations includes zeroing, negating, convolving, transposing, applying activation functions, and more. This setup generates semantically consistent images with unique and diverse distortions Figure 3. Although our set of perturbations is designed with random operations, we show that DeepAugment still outperforms other methods on benchmarks such as ImageNet-C and ImageNet-R. We provide the pseudocode in Appendix C.

For our experiments, we specifically use the CAE (Theis et al., 2017) and EDSR (Lim et al., 2017) architectures as the basis for DeepAugment. CAE is an autoencoder architecture, and EDSR is a superresolution architecture. These two architectures show the DeepAugment approach works with different architectures. Each clean image in the original dataset and passed through the network and is thereby stochastically distored, resulting in two distorted versions of the clean dataset (one for CAE and one for EDSR). We then train on the augmented and clean data simultaneously and call this approach DeepAugment.

|  | ImageNet-200 (%) | ImageNet-R (%) | Gap |
|---|---|---|---|
| ResNet-50 | 7.9 | 63.9 | 56.0 |
| + ImageNet-21K *Pretraining* (10× labeled data) | 7.0 | 62.8 | 55.8 |
| + CBAM (*Self-Attention*) | 7.0 | 63.2 | 56.2 |
| + $\ell_\infty$ Adversarial Training | 25.1 | 68.6 | 43.5 |
| + Speckle Noise | 8.1 | 62.1 | 54.0 |
| + Style Transfer Augmentation | 8.9 | 58.5 | 49.6 |
| + AugMix | 7.1 | 58.9 | 51.8 |
| + DeepAugment | 7.5 | 57.8 | 50.3 |
| + DeepAugment + AugMix | 8.0 | 53.2 | 45.2 |
| ResNet-152 (*Larger Models*) | 6.8 | 58.7 | 51.9 |

Table 1: ImageNet-200 and ImageNet-R top-1 error rates. ImageNet-200 uses the same 200 classes as ImageNet-R. DeepAugment+AugMix improves over the baseline by over 10 percentage points. ImageNet-21K Pretraining tests *Pretraining* and CBAM tests *Self-Attention*. Style Transfer, AugMix, and DeepAugment test *Diverse Data Augmentation* in contrast to simpler noise augmentations such as $\ell_\infty$ Adversarial Noise and Speckle Noise. While there remains much room for improvement, results indicate that progress on ImageNet-R is tractable.

## 5 EXPERIMENTS

### 5.1 SETUP

In this section we briefly describe the evaluated models, pretraining techniques, self-attention mechanisms, data augmentation methods, and note various implementation details.

**Model Architectures and Sizes.**   Most experiments are evaluated on a standard ResNet-50 model (He et al., 2015). Model size evaluations use ResNets or ResNeXts (Xie et al., 2016) of varying sizes.

**Pretraining.**   For pretraining we use ImageNet-21K which contains approximately 21,000 classes and approximately 14 million labeled training images, or around 10× more labeled training data than ImageNet-1K. We tune Kolesnikov et al. (2019)'s ImageNet-21K model. We also use a large pretrained ResNeXt-101 model from Mahajan et al. (2018). This was pre-trained on on approximately 1 billion Instagram images with hashtag labels and fine-tuned on ImageNet-1K. This Weakly Supervised Learning (WSL) pretraining strategy uses approximately 1000× more labeled data.

**Self-Attention.**   When studying self-attention, we employ CBAM (Woo et al., 2018) and SE (Hu et al., 2018) modules, two forms of self-attention that help models learn spatially distant dependencies.

**Data Augmentation.**   We use Style Transfer, AugMix, and DeepAugment to analyze the *Diverse Data Augmentation* hypothesis, and we contrast their performance with simpler noise augmentations such as Speckle Noise and adversarial noise. Style transfer (Geirhos et al., 2019) uses a style transfer network to apply artwork styles to training images. We use AugMix (Hendrycks et al., 2020b) which randomly composes simple augmentation operations (e.g., translate, posterize, solarize). DeepAugment, introduced above, distorts the weights and feedforward passes of image-to-image models to generate image augmentations. Speckle Noise data augmentation muliplies each pixel by $(1 + x)$ with $x$ sampled from a normal distribution (Rusak et al., 2020; Hendrycks and Dietterich, 2019). We also consider adversarial training as a form of adaptive data augmentation and use the model from Wong et al. (2020) trained against $\ell_\infty$ perturbations of size $\varepsilon = 4/255$.

### 5.2 RESULTS

We now perform experiments on ImageNet-R, StreetView StoreFronts, DeepFashion Remixed. We also evaluate on ImageNet-C and compare and contrast it with real distribution shifts.

**ImageNet-R.**   Table 1 shows performance on ImageNet-R as well as on ImageNet-200 (the original ImageNet data restricted to ImageNet-R's 200 classes). This has several implications regarding the four method-specific hypotheses. *Pretraining* with ImageNet-21K (approximately 10× labeled data) hardly helps. Appendix A shows WSL pretraining can help, but Instagram has renditions, while ImageNet excludes them; hence we conclude comparable pretraining was ineffective. Notice

*Self-Attention* increases the IID/OOD gap. Compared to simpler data augmentation techniques such as Speckle Noise, the *Diverse Data Augmentation* techniques of Style Transfer, AugMix, and DeepAugment improve generalization. Note AugMix and DeepAugment improve in-distribution performance whereas Style transfer hurts it. Also, our new DeepAugment technique is the best standalone method with an error rate of 57.8%. Last, *Larger Models* reduce the IID/OOD gap.

Regarding the three more abstract hypotheses, biasing networks away from natural textures through diverse data augmentation improved performance, so we find support for the *Texture Bias* hypothesis. The IID/OOD generalization gap varies greatly which condtradicts *Only IID Accuracy Matters*. Finally, since ImageNet-R contains real-world examples, and since synthetic data augmentation helps on ImageNet-R, we now have clear evidence against the *Synthetic* $\implies$ *Real* hypothesis.

**StreetView StoreFronts.** In Table 2, we evaluate data augmentation methods on SVSF and find that all of the tested methods have mostly similar performance and that no method helps much on country shift, where error rates roughly double across the board. Here evaluation is limited to augmentations due to a 30 day retention window for each instantiation of the dataset. Images captured in France contain noticeably different architectural styles and storefront designs than those captured in US/Mexico/Canada; meanwhile, we are unable to find conspicuous and consistent indicators of the camera and year. This may explain the relative insensitivity of evaluated methods to the camera and year shifts. Overall *Diverse Data Augmentation* shows limited benefit, suggesting either that data augmentation primarily helps combat texture bias as with ImageNet-R, or that existing augmentations are not diverse enough to capture high-level semantic shifts such as building architecture.

| Network | IID | Hardware Old | Year 2017 | 2018 | Location France |
|---|---|---|---|---|---|
| ResNet-50 | 27.2 | 28.6 | 27.7 | 28.3 | 56.7 |
| + Speckle Noise | 28.5 | 29.5 | 29.2 | 29.5 | 57.4 |
| + Style Transfer | 29.9 | 31.3 | 30.2 | 31.2 | 59.3 |
| + DeepAugment | 30.5 | 31.2 | 30.2 | 31.3 | 59.1 |
| + AugMix | 26.6 | 28.0 | 26.5 | 27.7 | 55.4 |

Table 2: SVSF classification error rates. Networks are robust to some natural distribution shifts but are substantially more sensitive the geographic shift. Here *Diverse Data Augmentation* hardly helps.

| Network | IID | OOD | Size Small | Large | Occlusion Slight/None | Heavy | Viewpoint No Wear | Side/Back | Zoom Medium | Large |
|---|---|---|---|---|---|---|---|---|---|---|
| ResNet-50 | 77.6 | 55.1 | 39.4 | 73.0 | 51.5 | 41.2 | 50.5 | 63.2 | 48.7 | 73.3 |
| + ImageNet-21K *Pretraining* | 80.8 | 58.3 | 40.0 | 73.6 | 55.2 | 43.0 | 63.0 | 67.3 | 50.5 | 73.9 |
| + SE (*Self-Attention*) | 77.4 | 55.3 | 38.9 | 72.7 | 52.1 | 40.9 | 52.9 | 64.2 | 47.8 | 72.8 |
| + Random Erasure | 78.9 | 56.4 | 39.9 | 75.0 | 52.5 | 42.6 | 53.4 | 66.0 | 48.8 | 73.4 |
| + Speckle Noise | 78.9 | 55.8 | 38.4 | 74.0 | 52.6 | 40.8 | 55.7 | 63.8 | 47.8 | 73.6 |
| + Style Transfer | 80.2 | 57.1 | 37.6 | 76.5 | 54.6 | 43.2 | 58.4 | 65.1 | 49.2 | 72.5 |
| + DeepAugment | 79.7 | 56.3 | 38.3 | 74.5 | 52.6 | 42.8 | 54.6 | 65.5 | 49.5 | 72.7 |
| + AugMix | 80.4 | 57.3 | 39.4 | 74.8 | 55.3 | 42.8 | 57.3 | 66.6 | 49.0 | 73.1 |
| ResNet-152 (*Larger Models*) | 80.0 | 57.1 | 40.0 | 75.6 | 52.3 | 42.0 | 57.7 | 65.6 | 48.9 | 74.4 |

Table 3: DeepFashion Remixed results. Unlike the previous tables, higher is better since all values are mAP scores for this multi-label classification benchmark. The "OOD" column is the average of the row's rightmost eight OOD values. All techniques do little to close the IID/OOD generalization gap.

**DeepFashion Remixed.** Table 3 shows our experimental findings on DFR, in which all evaluated methods have an average OOD mAP that is close to the baseline. In fact, most OOD mAP increases track IID mAP increases. In general, DFR's size and occlusion shifts hurt performance the most. We also evaluate with Random Erasure augmentation, which deletes rectangles within the image, to simulate occlusion (Zhong et al., 2017). Random Erasure improved occlusion performance, but Style Transfer helped even more. Nothing substantially improved OOD performance beyond what is explained by IID performance, so here it would appear that *Only IID Accuracy Matters*. Our results do not provide clear evidence for the *Larger Models*, *Self-Attention*, *Diverse Data Augmentation*, and *Pretraining* hypotheses.

**ImageNet-C.** We now consider a previous robustness benchmark to reassess all seven hypotheses. We use the ImageNet-C dataset (Hendrycks and Dietterich, 2019) which applies 15 common image

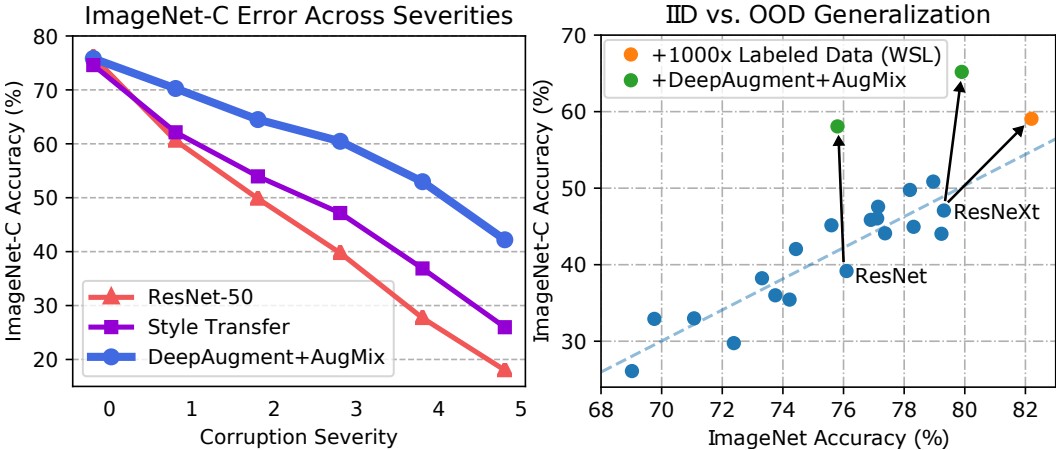

Figure 4: The left figure shows accuracy as a function of corruption severity. In the right figure, we show ImageNet accuracy and ImageNet-C accuracy. Previous architectural advances slowly translate to ImageNet-C performance improvements, but the DeepAugment+AugMix robustness intervention on a ResNet-50 approximately yields a 19% accuracy improvement.

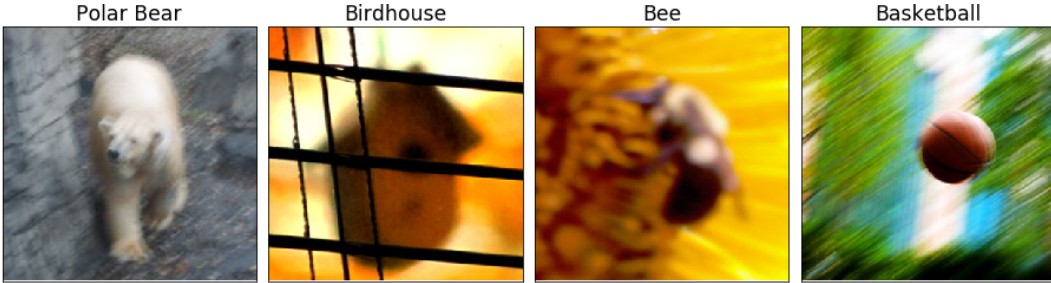

Figure 5: Examples of real-world blurry images from our collected dataset for ImageNet-C analysis.

corruptions (e.g., Gaussian noise, defocus blur, simulated fog, JPEG compression, etc.) across 5 severities to ImageNet-1K validation images. We find that DeepAugment improves robustness on ImageNet-C. Figure 4 shows that when models are trained with AugMix and DeepAugment, they attain the state-of-the-art, break the trendline, and exceed the corruption robustness provided by training on 1000× more labeled training data. Note the augmentations from AugMix and DeepAugment are disjoint from ImageNet-C's corruptions. Full results are shown in Appendix A's Table 8. This is evidence against the *Only IID Accuracy Matters* hypothesis and is evidence for the *Larger Models*, *Self-Attention*, *Diverse Data Augmentation*, *Pretraining*, and *Texture Bias* hypotheses.

Taori et al. (2020) remind us that ImageNet-C uses various *synthetic* corruptions and suggest that they are divorced from real-world robustness. Real-world robustness requires generalizing to naturally occurring corruptions such as snow, fog, blur, low-lighting noise, and so on, but it is an open question whether ImageNet-C's simulated corruptions meaningfully approximate real-world corruptions.

For our results analysis, we collect a small dataset of 1,000 real-world blurry images and find that ImageNet-C can track robustness to real-world corruptions. We collect the "Real Blurry Images" dataset with Flickr and query ImageNet object class names concatenated with the word "blurry." Examples are in Figure 5. We then evaluate various models on real-world blurry images and find that *all* the robustness interventions that help with ImageNet-C also help with real-world blurry images. Hence ImageNet-C can track performance on real-world corruptions. Moreover, DeepAugment+AugMix has the lowest error rate on Real Blurry Images, which again contradicts the *Synthetic ⇏ Real* hypothesis. The upshot is that ImageNet-C is a controlled and systematic proxy for real-world robustness.

We collect 1,000 blurry images to see whether improvements on ImageNet-C's simulated blurs correspond to improvements on real-world blurry images. Each image belongs to an ImageNet

| Hypothesis | ImageNet-C | Real Blurry Images | ImageNet-R | DFR | SVSF |
|---|---|---|---|---|---|
| *Larger Models* | + | + | + | − | |
| *Self-Attention* | + | + | − | − | |
| *Diverse Data Augmentation* | + | + | + | − | − |
| *Pretraining* | + | + | − | − | |

Table 4: A highly simplified account of each hypothesis when tested against different datasets. Evidence for is denoted "+", and "−" denotes an absence of evidence or evidence against.

class. Results from Table 5 show that *Larger Models*, *Self-Attention*, *Diverse Data Augmentation*, *Pretraining* all help, just like ImageNet-C. Here DeepAugment+AugMix attains state-of-the-art. These results suggest ImageNet-C's simulated corruptions track real-world corruptions. In hindsight, this is expected since various computer vision problems have used synthetic corruptions as proxies for real-world corruptions, for decades. In short, ImageNet-C is a diverse and systematic benchmark that is correlated with improvements on real-world corruptions.

## 6 CONCLUSION

In this paper we introduced three new benchmarks, ImageNet-Renditions, DeepFashion Remixed, and StreetView StoreFronts. With these benchmarks, we thoroughly tested seven robustness hypotheses– four about methods for robustness, and three about the nature of robustness.

Let us consider the first four hypotheses, using the new information from ImageNet-C and our three new benchmarks. The *Larger Models* hypothesis was supported with ImageNet-C and ImageNet-R, but not with DFR. While *Self-Attention* noticeably helped ImageNet-C, it did not help with ImageNet-R and DFR. *Diverse Data Augmentation* was ineffective for SVSF and DFR, but it greatly improved ImageNet-C and ImageNet-R accuracy. *Pretraining* greatly helped with ImageNet-C but hardly helped with DFR and ImageNet-R. This is summarized in Table 4. It was not obvious *a priori* that synthetic *Diverse Data Augmentation* could improve ImageNet-R accuracy, nor did previous research suggest that *Pretraining* would sometimes be ineffective. While no single method consistently helped across all distribution shifts, some helped more than others.

Our analysis of these four hypotheses have implications for the remaining three hypotheses. Regarding *Texture Bias*, ImageNet-R shows that networks do not generalize well to renditions (which have different textures), but that diverse data augmentation (which often distorts textures) can recover accuracy. More generally, larger models and diverse data augmentation consistently helped on ImageNet-R, ImageNet-C, and Blurry Images, suggesting that these two interventions reduce texture bias. However, these methods helped little for geographic shifts, showing that there is more to robustness than texture bias alone. Regarding *Only IID Accuracy Matters*, while IID accuracy is a strong predictor of OOD accuracy, it is not decisive—Table 4 shows that many methods improve robustness across multiple distribution shifts, and recent experiments in NLP provide further counterexamples (Hendrycks et al., 2020a). Finally, *Synthetic* $\not\Rightarrow$ *Real* has clear counterexamples given that DeepAugment greatly increases accuracy on ImageNet-R and Real Blurry Images. In summary, some previous hypotheses are implausible, and the Texture Bias hypothesis has the most support.

Our seven hypotheses presented several conflicting accounts of robustness. What led to this conflict? We suspect it is because robustness is not one scalar like accuracy. The research community is reasonable in judging IID accuracy with a *univariate* metric like ImageNet classification accuracy, as models with higher ImageNet accuracy reliably have better fine-tuned classification accuracy on other tasks (Kornblith et al., 2018). In contrast, we argue it is too simplistic to judge OOD accuracy with a univariate metric like, say, ImageNetV2 or ImageNet-C accuracy. Instead we hypothesize that robustness is multivariate. This *Multivariate* hypothesis means that there is not a single scalar model property that wholly governs natural model robustness.

If robustness has many faces, future work should evaluate robustness using many distribution shifts; for example, ImageNet models should at least be tested against ImageNet-C and ImageNet-R. Future work could further characterize the space of distribution shifts. However, due to this paper, there are now more out-of-distribution robustness datasets than there are published robustness methods. Hence the research community should prioritize creating new robustness methods. If our *Multivariate* hypothesis is true, research should shift toward using multiple tests to develop models that are both robust and safe.

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

## A    ADDITIONAL RESULTS

**ImageNet-R.**    Expanded ImageNet-R results are in Table 7.

WSL pretraining on Instagram images appears to yield dramatic improvements on ImageNet-R, but the authors note the prevalence of artistic renditions of object classes on the Instagram platform. While ImageNet's data collection process actively excluded renditions, we do not have reason to believe the Instagram dataset excluded renditions. On a ResNeXt-101 32×8d model, WSL pretraining improves ImageNet-R performance by a massive 37.5% from 57.5% top-1 error to 24.2%. Ultimately, without examining the training images we are unable to determine whether ImageNet-R represents an actual distribution shift to the Instagram WSL models. However, we also observe that with greater controls, that is with ImageNet-21K pre-training, pretraining hardly helped ImageNet-R performance, so it is not clear that more pretraining data improves ImageNet-R performance.

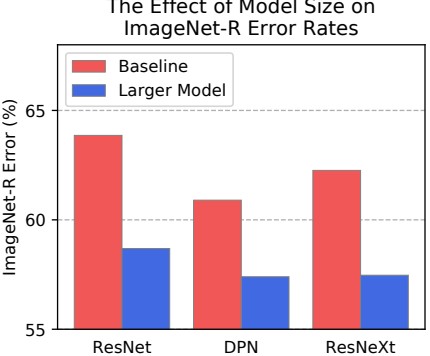

Figure 6: Larger models improve robustness on ImageNet-R. The baseline models are ResNet-50, DPN-68, and ResNeXt-50 (32 × 4d). The larger models are ResNet-152, DPN-98, and ResNeXt-101 (32 × 8d). The baseline ResNeXt has a 7.1% ImageNet error rate, while the large has a 6.2% error rate.

Increasing model size appears to automatically improve ImageNet-R performance, as shown in Figure 6. A ResNet-50 (25.5M parameters) has 63.9% error, while a ResNet-152 (60M) has 58.7% error. ResNeXt-50 32×4d (25.0M) attains 62.3% error and ResNeXt-101 32×8d (88M) attains 57.5% error.

**ImageNet-C.**    Expanded ImageNet-C results are Table 8. We also tested whether model size improves performance on ImageNet-C for even larger models. With a different codebase, we trained ResNet-50, ResNet-152, and ResNet-500 models which achieved 80.6, 74.0, and 68.5 mCE respectively.

Expanded comparisons between ImageNet-C and Real Blurry Images is in Table 5.

| Network | Defocus Blur | Glass Blur | Motion Blur | Zoom Blur | ImageNet-C Blur Mean | Real Blurry Images |
|---|---|---|---|---|---|---|
| ResNet-50 | 61 | 73 | 61 | 64 | 65 | 58.7 |
| + ImageNet-21K *Pretraining* | 56 | 69 | 53 | 59 | 59 | 54.8 |
| + CBAM (*Self-Attention*) | 60 | 69 | 56 | 61 | 62 | 56.5 |
| + $\ell_\infty$ Adversarial Training | 80 | 71 | 72 | 71 | 74 | 71.6 |
| + Speckle Noise | 57 | 68 | 60 | 64 | 62 | 56.9 |
| + Style Transfer | 57 | 68 | 55 | 64 | 61 | 56.7 |
| + AugMix | 52 | 65 | 46 | 51 | 54 | 54.4 |
| + DeepAugment | 48 | 60 | 51 | 61 | 55 | 54.2 |
| + DeepAugment+AugMix | 41 | 53 | 39 | 48 | 45 | 51.7 |
| ResNet-152 (*Larger Models*) | 67 | 81 | 66 | 74 | 58 | 54.3 |

Table 5: ImageNet-C Blurs (Defocus, Glass, Motion, Zoom) vs Real Blurry Images. All values are error rates and percentages. The rank orderings of the models on Real Blurry Images are similar to the rank orderings for "ImageNet-C Blur Mean," so ImageNet-C's simulated blurs track real-world blur performance.

**ImageNet-A.**    ImageNet-A (Hendrycks et al., 2019b) is an adversarially filtered test set and is constructed based on existing model weaknesses (see (Wang et al., 2020) for another robustness dataset algorithmically determined by model weaknesses). This dataset contains examples that are difficult for a ResNet-50 to classify, so examples solvable by simple spurious cues are are especially infrequent in this dataset. Results are in Table 9. Notice Res2Net architectures (Gao et al., 2019b) can greatly improve accuracy. Results also show that *Larger Models*, *Self-Attention*, and *Pretraining* help, while *Diverse Data Augmentation* usually does not help substantially.

**Implications for the Four Method Hypotheses.**

The *Larger Models* hypothesis has support with ImageNet-C (+), ImageNet-A (+), ImageNet-R (+), yet does not markedly improve DFR (−) performance.

The *Self-Attention* hypothesis has support with ImageNet-C (+), ImageNet-A (+), yet does not help ImageNet-R (−) and DFR (−) performance.

The *Diverse Data Augmentation* hypothesis has support with ImageNet-C (+), ImageNet-R (+), yet does not markedly improve ImageNet-A (−), DFR(−), nor SVSF (−) performance.

The *Pretraining* hypothesis has support with ImageNet-C (+), ImageNet-A (+), yet does not markedly improve DFR (−) nor ImageNet-R (−) performance.

| Hypothesis | ImageNet-C | ImageNet-A | ImageNet-R | DFR | SVSF |
|---|---|---|---|---|---|
| *Larger Models* | + | + | + | − | |
| *Self-Attention* | + | + | − | − | |
| *Diverse Data Augmentation* | + | − | + | − | − |
| *Pretraining* | + | + | − | − | |

Table 6: A highly simplified account of each hypothesis when tested against different datasets. This table includes ImageNet-A results.

|  | ImageNet-200 (%) | ImageNet-R (%) | Gap |
|---|---|---|---|
| ResNet-50 (He et al., 2015) | 7.9 | 63.9 | 56.0 |
| + ImageNet-21K *Pretraining* (10× data) | 7.0 | 62.8 | 55.8 |
| + CBAM (*Self-Attention*) | 7.0 | 63.2 | 56.2 |
| + $\ell_\infty$ Adversarial Training | 25.1 | 68.6 | 43.5 |
| + Speckle Noise | 8.1 | 62.1 | 54.0 |
| + Style Transfer | 8.9 | 58.5 | 49.6 |
| + AugMix | 7.1 | 58.9 | 51.8 |
| + DeepAugment | 7.5 | 57.8 | 50.3 |
| + DeepAugment + AugMix | 8.0 | 53.2 | 45.2 |
| ResNet-101 (*Larger Models*) | 7.1 | 60.7 | 53.6 |
| + SE (*Self-Attention*) | 6.7 | 61.0 | 54.3 |
| ResNet-152 (*Larger Models*) | 6.8 | 58.7 | 51.9 |
| + SE (*Self-Attention*) | 6.6 | 60.0 | 53.4 |
| ResNeXt-101 32×4d (*Larger Models*) | 6.8 | 58.0 | 51.2 |
| + SE (*Self-Attention*) | 5.9 | 59.6 | 53.7 |
| ResNeXt-101 32×8d (*Larger Models*) | 6.2 | 57.5 | 51.3 |
| + WSL *Pretraining* (1000× data) | 4.1 | 24.2 | 20.1 |
| + DeepAugment + AugMix | 6.1 | 47.9 | 41.8 |

Table 7: ImageNet-200 and ImageNet-Renditions error rates. ImageNet-21K and WSL Pretraining test the *Pretraining* hypothesis, and here pretraining gives mixed benefits. CBAM and SE test the *Self-Attention* hypothesis, and these *hurt* robustness. ResNet-152 and ResNeXt-101 32×8d test the *Larger Models* hypothesis, and these help. Other methods augment data, and Style Transfer, AugMix, and DeepAugment provide support for the *Diverse Data Augmentation* hypothesis.

|  | Clean | mCE | Noise | | | Blur | | | | Weather | | | | Digital | | | |
|---|---|---|---|---|---|---|---|---|---|---|---|---|---|---|---|---|---|
|  |  |  | Gauss. | Shot | Impulse | Defocus | Glass | Motion | Zoom | Snow | Frost | Fog | Bright | Contrast | Elastic | Pixel | JPEG |
| ResNet-50 | 23.9 | 76.7 | 80 | 82 | 83 | 75 | 89 | 78 | 80 | 78 | 75 | 66 | 57 | 71 | 85 | 77 | 77 |
| + ImageNet-21K *Pretraining* | 22.4 | 65.8 | 61 | 64 | 63 | 69 | 84 | 68 | 74 | 69 | 71 | 61 | 53 | 53 | 81 | 54 | 63 |
| + SE (*Self-Attention*) | 22.4 | 68.2 | 63 | 66 | 66 | 71 | 82 | 67 | 74 | 74 | 72 | 64 | 55 | 71 | 73 | 60 | 67 |
| + CBAM (*Self-Attention*) | 22.4 | 70.0 | 67 | 68 | 68 | 74 | 83 | 71 | 76 | 73 | 72 | 65 | 54 | 70 | 79 | 62 | 67 |
| + $\ell_\infty$ Adversarial Training | 46.2 | 94.0 | 91 | 92 | 95 | 97 | 86 | 92 | 88 | 93 | 99 | 118 | 104 | 111 | 90 | 72 | 81 |
| + Speckle Noise | 24.2 | 68.3 | 51 | 47 | 55 | 70 | 83 | 77 | 80 | 76 | 71 | 66 | 57 | 70 | 82 | 72 | 69 |
| + Style Transfer | 25.4 | 69.3 | 66 | 67 | 68 | 70 | 82 | 69 | 80 | 68 | 71 | 65 | 58 | 66 | 78 | 62 | 70 |
| + AugMix | 22.5 | 65.3 | 67 | 66 | 68 | 64 | 79 | 59 | 64 | 69 | 68 | 65 | 54 | 57 | 74 | 60 | 66 |
| + DeepAugment | 23.3 | 60.4 | 49 | 50 | 47 | 59 | 73 | 65 | 76 | 64 | 60 | 58 | 51 | 61 | 76 | 48 | 67 |
| + DeepAugment + AugMix | 24.2 | 53.6 | 46 | 45 | 44 | 50 | 64 | 50 | 61 | 58 | 57 | 54 | 52 | 48 | 71 | 43 | 61 |
| ResNet-152 (*Larger Models*) | 21.7 | 69.3 | 73 | 73 | 76 | 67 | 81 | 66 | 74 | 71 | 68 | 62 | 51 | 67 | 76 | 69 | 65 |
| ResNeXt-101 32×8d (*Larger Models*) | 20.7 | 66.7 | 68 | 69 | 71 | 65 | 79 | 66 | 71 | 69 | 66 | 60 | 50 | 66 | 74 | 61 | 64 |
| + WSL *Pretraining* (1000× data) | 17.8 | 51.7 | 49 | 50 | 51 | 53 | 72 | 55 | 63 | 53 | 51 | 42 | 37 | 41 | 67 | 40 | 51 |
| + DeepAugment + AugMix | 20.1 | 44.5 | 36 | 35 | 34 | 43 | 55 | 42 | 55 | 48 | 48 | 47 | 43 | 39 | 59 | 34 | 50 |

Table 8: Clean Error, Corruption Error (CE), and mean CE (mCE) values for various models and training methods on ImageNet-C. The mCE value is computed by averaging across all 15 CE values. A CE value greater than 100 (e.g. adversarial training on contrast) denotes worse performance than AlexNet. DeepAugment+AugMix improves robustness by over 23 mCE.

|  | ImageNet-A (%) |
|---|---|
| ResNet-50 | 2.2 |
| + ImageNet-21K *Pretraining* (10× data) | 11.4 |
| + Squeeze-and-Excitation (*Self-Attention*) | 6.2 |
| + CBAM (*Self-Attention*) | 6.9 |
| + $\ell_\infty$ Adversarial Training | 1.7 |
| + Style Transfer | 2.0 |
| + AugMix | 3.8 |
| + DeepAugment | 3.5 |
| + DeepAugment + AugMix | 3.9 |
| ResNet-152 (*Larger Models*) | 6.1 |
| ResNet-152+Squeeze-and-Excitation (*Self-Attention*) | 9.4 |
| Res2Net-50 v1b | 14.6 |
| Res2Net-152 v1b (*Larger Models*) | 22.4 |
| ResNeXt-101 (32 × 8d) (*Larger Models*) | 10.2 |
| + WSL *Pretraining* (1000× data) | 45.4 |
| + DeepAugment + AugMix | 11.5 |

Table 9: ImageNet-A top-1 accuracy.

## B  FURTHER DATASET DESCRIPTIONS

**ImageNet-R Classes.**  The 200 ImageNet classes and their WordNet IDs in ImageNet-R are as follows.

Goldfish, great white shark, hammerhead, stingray, hen, ostrich, goldfinch, junco, bald eagle, vulture, newt, axolotl, tree frog, iguana, African chameleon, cobra, scorpion, tarantula, centipede, peacock, lorikeet, hummingbird, toucan, duck, goose, black swan, koala, jellyfish, snail, lobster, hermit crab, flamingo, american egret, pelican, king penguin, grey whale, killer whale, sea lion, chihuahua, shih tzu, afghan hound, basset hound, beagle, bloodhound, italian greyhound, whippet, weimaraner, yorkshire terrier, boston terrier, scottish terrier, west highland white terrier, golden retriever, labrador retriever, cocker spaniels, collie, border collie, rottweiler, german shepherd dog, boxer, french bulldog, saint bernard, husky, dalmatian, pug, pomeranian, chow chow, pembroke welsh corgi, toy poodle, standard poodle, timber wolf, hyena, red fox, tabby cat, leopard, snow leopard, lion, tiger, cheetah, polar bear, meerkat, ladybug, fly, bee, ant, grasshopper, cockroach, mantis, dragonfly, monarch butterfly, starfish, wood rabbit, porcupine, fox squirrel, beaver, guinea pig, zebra, pig, hippopotamus, bison, gazelle, llama, skunk, badger, orangutan, gorilla, chimpanzee, gibbon, baboon, panda, eel, clown fish, puffer fish, accordion, ambulance, assault rifle, backpack, barn, wheelbarrow, basketball, bathtub, lighthouse, beer glass, binoculars, birdhouse, bow tie, broom, bucket, cauldron, candle, cannon, canoe, carousel, castle, mobile phone, cowboy hat, electric guitar, fire engine, flute, gasmask, grand piano, guillotine, hammer, harmonica, harp, hatchet, jeep, joystick, lab coat, lawn mower, lipstick, mailbox, missile, mitten, parachute, pickup truck, pirate ship, revolver, rugby ball, sandal, saxophone, school bus, schooner, shield, soccer ball, space shuttle, spider web, steam locomotive, scarf, submarine, tank, tennis ball, tractor, trombone, vase, violin, military aircraft, wine bottle, ice cream, bagel, pretzel, cheeseburger, hotdog, cabbage, broccoli, cucumber, bell pepper, mushroom, Granny Smith, strawberry, lemon, pineapple, banana, pomegranate, pizza, burrito, espresso, volcano, baseball player, scuba diver, acorn.

n01443537, n01484850, n01494475, n01498041, n01514859, n01518878, n01531178, n01534433, n01614925, n01616318, n01630670, n01632777, n01644373, n01677366, n01694178, n01748264, n01770393, n01774750, n01784675, n01806143, n01820546, n01833805, n01843383, n01847000, n01855672, n01860187, n01882714, n01910747, n01944390, n01983481, n01986214, n02007558, n02009912, n02051845, n02056570,

n02066245, n02071294, n02077923, n02085620, n02086240, n02088094, n02088238,
n02088364, n02088466, n02091032, n02091134, n02092339, n02094433, n02096585,
n02097298, n02098286, n02099601, n02099712, n02102318, n02106030, n02106166,
n02106550, n02106662, n02108089, n02108915, n02109525, n02110185, n02110341,
n02110958, n02112018, n02112137, n02113023, n02113624, n02113799, n02114367,
n02117135, n02119022, n02123045, n02128385, n02128757, n02129165, n02129604,
n02130308, n02134084, n02138441, n02165456, n02190166, n02206856, n02219486,
n02226429, n02233338, n02236044, n02268443, n02279972, n02317335, n02325366,
n02346627, n02356798, n02363005, n02364673, n02391049, n02395406, n02398521,
n02410509, n02423022, n02437616, n02445715, n02447366, n02480495, n02480855,
n02481823, n02483362, n02486410, n02510455, n02526121, n02607072, n02655020,
n02672831, n02701002, n02749479, n02769748, n02793495, n02797295, n02802426,
n02808440, n02814860, n02823750, n02841315, n02843684, n02883205, n02906734,
n02909870, n02939185, n02948072, n02950826, n02951358, n02966193, n02980441,
n02992529, n03124170, n03272010, n03345487, n03372029, n03424325, n03452741,
n03467068, n03481172, n03494278, n03495258, n03498962, n03594945, n03602883,
n03630383, n03649909, n03676483, n03710193, n03773504, n03775071, n03888257,
n03930630, n03947888, n04086273, n04118538, n04133789, n04141076, n04146614,
n04147183, n04192698, n04254680, n04266014, n04275548, n04310018, n04325704,
n04347754, n04389033, n04409515, n04465501, n04487394, n04522168, n04536866,
n04552348, n04591713, n07614500, n07693725, n07695742, n07697313, n07697537,
n07714571, n07714990, n07718472, n07720875, n07734744, n07742313, n07745940,
n07749582, n07753275, n07753592, n07768694, n07873807, n07880968, n07920052,
n09472597, n09835506, n10565667, n12267677.

**SVSF.** The classes are

- auto shop
- bakery
- bank
- beauty salon
- car dealer
- car wash
- cell phone store
- dentist
- discount store
- dry cleaner
- furniture store
- gas station
- gym
- hardware store
- hotel
- liquor store
- pharmacy
- religious institution
- storage facility
- veterinary care.

**DeepFashion Remixed.** The classes are

- short sleeve top
- long sleeve top
- short sleeve outerwear
- long sleeve outerwear
- vest
- sling
- shorts
- trousers
- skirt
- short sleeve dress
- long sleep dress
- vest dress
- sling dress.

Size (small, moderate, or large) defines how much of the image the article of clothing takes up. Occlusion (slight, medium, or heavy) defines the degree to which the object is occluded from the camera. Viewpoint (front, side/back, or not worn) defines the camera position relative to the article of clothing. Zoom (no zoom, medium, or large) defines how much camera zoom was used to take the picture.

|  | Represented Distribution Shifts |
|---|---|
| ImageNet-Renditions | artistic renditions (cartoons, graffiti, embroidery, graphics, origami, paintings, sculptures, sketches, tattoos, toys, ...) |
| DeepFashion Remixed | occlusion, size, viewpoint, zoom |
| StreetView StoreFronts | camera, capture year, country |

Table 10: Various distribution shifts represented in our three new benchmarks. ImageNet-Renditions is a new test set for ImageNet trained models measuring robustness to various object renditions. DeepFashion Remixed and StreetView StoreFronts each contain a training set and multiple test sets capturing a variety of distribution shifts.

|  | Training set | Testing images |
|---|---|---|
| ImageNet-R | 1281167 | 30000 |
| DFR | 48000 | 42640, 7440, 28160, 10360, 480, 11040, 10520, 10640 |
| SVSF | 200000 | 10000, 10000, 10000, 8195, 9788 |

Table 11: Number of images in each training and test set. ImageNet-R training set refers to the ILSVRC 2012 training set (Deng et al., 2009). DeepFashion Remixed test sets are: in-distribution, occlusion - none/slight, occlusion - heavy, size - small, size - large, viewpoint - frontal, viewpoint - not-worn, zoom-in - medium, zoom-in - large. StreetView StoreFronts test sets are: in-distribution, capture year - 2018, capture year - 2017, camera system - new, country - France.

## C   DEEPAUGMENT DETAILS

**Pseudocode.**   Below is Pythonic pseudocode for DeepAugment. The basic structure of DeepAugment is agnostic to the backbone network used, but specifics such as which layers are chosen for various transforms may vary as the backbone architecture varies. We do not need to train many different image-to-image models to get diverse distortions (Zhang et al., 2018; Lee et al., 2020). We only use two existing models, the EDSR super-resolution model (Lim et al., 2017) and the CAE image compression model (Theis et al., 2017). See full code for such details.

At a high level, DeepAugment processes each image with an image-to-image network. The image-to-image network's weights and feedforward activations are distorted with each pass. The distortion is made possible by, for example, negating the network's weights and applying dropout to the feedforward activations. These modifications were not carefully chosen and demonstrate the utility of mixing together diverse operations without tuning. The resulting image is distorted and saved. This process generates an augmented dataset.

```python
1  def main():
2      net.apply_weights(deepAugment_getNetwork())  # EDSR, CAE, ...
3      for image in dataset:  # May be the ImageNet training set
4          if np.random.uniform() < 0.05:  # Arbitrary refresh prob
5              net.apply_weights(deepAugment_getNetwork())
6          new_image = net.deepAugment_forwardPass(image)
7
8  def deepAugment_getNetwork():
9      weights = load_clean_weights()
10     weight_distortions = sample_weight_distortions()
11     for d in weight_distortions:
12         weights = apply_distortion(d, weights)
13     return weights
14
15 def sample_weight_distortions():
16     distortions = [
17         negate_weights,
18         zero_weights,
19         flip_transpose_weights,
20         ...
21     ]
22
23     return random_subset(distortions)
24
25 def sample_signal_distortions():
26     distortions = [
27         gelu,
28         negate_signal_random_mask,
29         flip_signal,
30         ...
31     ]
32
33     return random_subset(distortions)
34
35
36 class Network():
37     def apply_weights(weights):
38         ... # Apply given weight tensors to network
39
40     # Clean forward pass. Compare to deepAugment_forwardPass()
41     def clean_forwardPass(X):
42         X = network.block1(X)
43         X = network.block2(X)
44         ...
45         X = network.blockN(X)
46         return X
47
48     # Our forward pass. Compare to clean_forwardPass()
49     def deepAugment_forwardPass(X):
50         # Returns a list of distortions, each of which
51         # will be applied at a different layer.
52         signal_distortions = sample_signal_distortions()
53
54         X = network.block1(X)
55         apply_layer_1_distortions(X, signal_distortions)
56         X = network.block2(X)
57         apply_layer_2_distortions(X, signal_distortions)
58         ...
59         apply_layer_N-1_distortions(X, signal_distortions)
60         X = network.blockN(X)
61         apply_layer_N_distortions(X, signal_distortions)
62
63     return X
```

**Ablations.** We run ablations on DeepAugment to understand the contributions from the EDSR and CAE models independently. Table 13 contains results of these experiments on ImageNet-R and Table 12 contains results of these experiments on ImageNet-C. In both tables, "DeepAugment (EDSR)" and "DeepAugment (CAE)" refer to experiments where we only use a single extra augmented training set (+ the standard training set), and train on those images.

| | | | Noise | | | Blur | | | | Weather | | | | Digital | | |
|---|---|---|---|---|---|---|---|---|---|---|---|---|---|---|---|---|
| | *Clean* | *mCE* | Gauss. | Shot | Impulse | Defocus | Glass | Motion | Zoom | Snow | Frost | Fog | Bright | Contrast | Elastic | Pixel | JPEG |
| ResNet-50 | 23.9 | 76.7 | 80 | 82 | 83 | 75 | 89 | 78 | 80 | 78 | 75 | 66 | 57 | 71 | 85 | 77 | 77 |
| + DeepAugment (EDSR) | 23.5 | 64.0 | 56 | 57 | 54 | 64 | 77 | 71 | 78 | 68 | 64 | 64 | 55 | 64 | 78 | 46 | 67 |
| + DeepAugment (CAE) | 23.2 | 67.0 | 58 | 60 | 62 | 62 | 75 | 73 | 77 | 68 | 66 | 60 | 52 | 66 | 80 | 63 | 78 |
| + DeepAugment (Both) | 23.3 | 60.4 | 49 | 50 | 47 | 59 | 73 | 65 | 76 | 64 | 60 | 58 | 51 | 61 | 76 | 48 | 67 |

Table 12: Clean Error, Corruption Error (CE), and mean CE (mCE) values for DeepAugment ablations on ImageNet-C. The mCE value is computed by averaging across all 15 CE values.

**Noise2Net.** We show that untrained, randomly sampled neural networks can provide useful deep augmentations, highlighting the efficacy of the DeepAugment approach. While in the main paper we use EDSR and CAE to create DeepAugment augmentations, in this section we explore the use of randomly initialized image-to-image networks to generate diverse image augmentations. We propose a DeepAugment method, *Noise2Net*.

In Noise2Net, the architecture and weights are randomly sampled. Noise2Net is the composition of several residual blocks: $Block(x) = x + \varepsilon \cdot f_\Theta(x)$, where $\Theta$ is randomly initialized and $\varepsilon$ is a parameter that controls the strength of the augmentation. For all our experiments, we use 4 Res2Net blocks (Gao et al., 2019a) and $\varepsilon \sim U(0.375, 0.75)$. The weights of Noise2Net are resampled at every minibatch, and the dilation and kernel sizes of all the convolutions used in Noise2Net are randomly sampled every epoch. Hence Noise2Net augments an image to an augmented image by processing the image through a randomly sampled network with random weights.

Recall that in the case of EDSR and CAE, we used networks to generate a static dataset, and then we trained normally on that static dataset. This setup could not be done on-the-fly. That is because we fed in one example at a time with EDSR and CAE. If we pass the entire minibatch through EDSR or CAE, we will end up applying the same augmentation to all images in the minibatch, reducing stochasticity and augmentation diversity. In contrast, Noise2Net enables us to process batches of images on-the-fly and obviates the need for creating a static augmented dataset.

In Noise2Net, each example is processed differently in parallel, so we generate more diverse augmentations in real-time. To make this possible, we use grouped convolutions. A grouped convolution with number of groups $= N$ will take a set of $kN$ channels as input, and apply $N$ independent convolutions on channels $\{1, \ldots, k\}, \{k+1, \ldots, 2k\}, \ldots, \{(N-1)k+1, \ldots, Nk\}$. Given a minibatch of size $B$, we can apply a randomly initialized grouped convolution with $N = B$ groups in order to apply a different random convolutional filter to each element in the batch in a single forward pass. By replacing all the convolutions in each Res2Net block with a grouped convolution and randomly initializing network weights, we arrive at Noise2Net, a variant of DeepAugment. See Figure 7 for a high-level overview of Noise2Net and Figure 8 for sample outputs.

We evaluate the Noise2Net variant of DeepAugment on ImageNet-R. Table 13 shows that it outperforms the EDSR and CAE variants of DeepAugment, even though the network architecture is randomly sampled, its weights are random, and the network is not trained. This demonstrates the flexibility of the DeepAugment approach. Below is Pythonic pseudocode for training a classifier using the Noise2Net variant of DeepAugment.

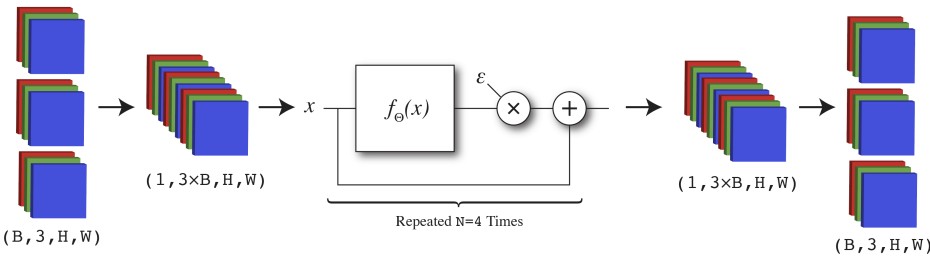

Figure 7: Parallel augmentation with Noise2Net. We collapse batches to the channel dimension to ensure that different transformations are applied to each image in the batch. Feeding images into the network in the standard way would result in the same augmentation being applied to each image, which is undesirable. The function $f_\Theta(x)$ is a Res2Net block with all convolutions replaced with grouped convolutions.

|  | ImageNet-200 (%) | ImageNet-R (%) | Gap |
|---|---|---|---|
| ResNet-50 | 7.9 | 63.9 | 56.0 |
| + DeepAugment (EDSR) | 7.9 | 60.3 | 55.1 |
| + DeepAugment (CAE) | 7.6 | 58.5 | 50.9 |
| + DeepAugment (EDSR + CAE) | 7.5 | 57.8 | 50.3 |
| + DeepAugment (Noise2Net) | 7.2 | 57.6 | 50.4 |
| + DeepAugment (All 3) | 7.4 | 56.0 | 48.6 |

Table 13: DeepAugment ablations on ImageNet-200 and ImageNet-Renditions.

```python
def train_one_epoch(classifier, batch_size, dataloader):
    noise2net = Noise2Net(batch_size=batch_size)
    for batch, target in dataloader:
        noise2net.reload_weights()
        noise2net.set_epsilon(random.uniform(0.375, 0.75))
        logits = model(noise2net.forward(batch))
        ... # Calculate loss and backrop

def train():
    for epoch in range(epochs):
        train_one_epoch(classifier, batch_size, dataloader)

class Noise2Net:
    def __init__(self, batch_size=5):
        self.block1 = Res2NetBlock(batch_size=batch_size)
        self.block2 = Res2NetBlock(batch_size=batch_size)
        self.block3 = Res2NetBlock(batch_size=batch_size)
        self.block4 = Res2NetBlock(batch_size=batch_size)

    def reload_weights(self):
        ... # Reload Network parameters

    def set_epsilon(self, new_eps):
        self.epsilon = new_eps

    def forward(self, x):
        x = x + self.block1(x) * self.epsilon
        x = x + self.block2(x) * self.epsilon
        x = x + self.block3(x) * self.epsilon
        x = x + self.block4(x) * self.epsilon
        return x
```

$\varepsilon = 0.00$     $\varepsilon = 0.25$     $\varepsilon = 0.75$

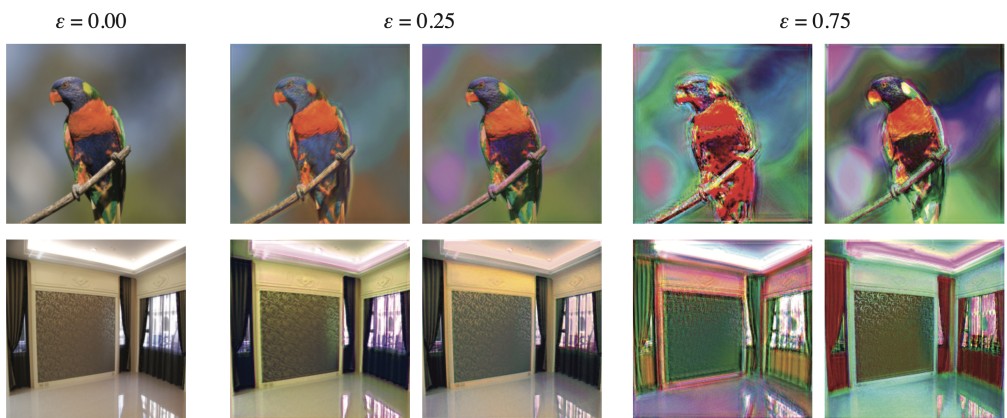

Figure 8: Example outputs of Noise2Net for different values of $\varepsilon$. Note $\varepsilon = 0$ is the original image.

