# OpenReview forum: "A Critical Analysis of Distribution Shift"
_ICLR.cc/2021/Conference — Reject_

### Official Review · AnonReviewer4 · 2020-10-28
**Solid work!**

**Rating:** 8
**Confidence:** 4

**Review:**

Summary: This paper investigates the robustness problem of computer vision model. To study the model robustness in a controlled setting, the author introduces three new robustness benchmarks: ImageNet-R, StreetView StoreFronts and DeepFashion Remixed. Each of them address different aspects of distribution drift in the real world. The author evaluates seven popular hypotheses on model robustness in the community on the three new datasets and has found counter-example for most of them. Based on those new results, the author concluded that model robustness problem is multi-variate in nature: no single solution could handle all aspects yet. And future work should be tested on multiple datasets to prove robustness. Moreover, the author also proposes a new data augmentation method using perturbed image-to-image deep learning model to generate visually diverse augmentations.


Significance: This paper is a solid work on the robustness problem. It systematically evaluated common hypotheses and successfully found counter-example on all of them except Texture Bias. The analysis is insightful and supported by the experimental results. The authors also provides three new carefully designed datasets for future work evaluation. While the study of using deep neural network to generate training image is not new, DeepAugmentation is still an innovative and practical way for data augmentation purpose.

Question: On DeepFashion Remixed datasets, it seems large zoom has better result than medium zoom. Is there a good explanation for that, considering the original image has no zoom-in?

Clarity: The author did a great job on explaining the idea, objective and approach.

---

> ### Author Response · Authors · 2020-11-17
> **DeepFashion Remixed Clarification**
>
> Thank you for your careful analysis of our paper and your positive review!
>
> “On DeepFashion Remixed datasets, it seems large zoom has better result than medium zoom. Is there a good explanation for that, considering the original image has no zoom-in?”
> We suspect this is because images with large zoom in usually have only one object visible per image rather than many object per image. This makes the multi-label ground truth easier to predict.

---

> ### Comment · Area_Chair1 · 2020-11-22
> **R4, what do you think of the other reviews?**
>
> Greetings R4,
>
> Can you take a look at the other reviews and author responses and let us know what you think.
>
> Thanks

---

> > ### Comment · AnonReviewer4 · 2020-11-24
> > **I continue to hold my previous rating of clear accept**
> >
> > Thanks to the authors for the explanations

---

### Official Review · AnonReviewer2 · 2020-10-28

**Rating:** 5
**Confidence:** 4

**Review:**

This paper provides a empirical study on the robustness of image classification models to distributions shifts. The authors construct three benchmark datasets that control for effects like artistic renditions of common classes, view-point changes, and geographic shifts (among others). The datasets are then used to test various hypotheses regarding robustness enhancing measures empirically. The authors additionally propose a novel augmentation scheme, that uses deep image processing networks together with random perturbations of their weights to synthesize distorted image samples.

---- Strengths ----

The paper tackles and important topic. I agree with the authors that robustness is "multivariate", i.e. can not be improved by a single factor. The paper makes an effort to disentangle various factors and test them in isolation.

The paper provides evidence that ImageNet-C can be used to make conclusion about real behavior of models, despite being based on synthetic image transformations.

The paper is well written and provides comprehensive experiments.

--- Weaknesses ---

The conclusions that result from the empirical findings are unfortunately not very crisp. Some hypotheses are supported by some datasets, others are not. No clear conclusions that hold across datasets can be drawn. It seems that we can't learn much from the experiments, and that the answer  to the questions "What improves robustness?" is still very much "it depends on what you are testing on". This is deeply unsatisfying, as true robustness  actually should not depend on the dataset. Table 4 provides a simplified summary of various hypotheses and how they are supported by different datasets. This table doesn't look too surprising: Why should, for example, self-attention help to improve performance on artistic renditions, something that  is very different from a blurred variant of an image? Why would it help classify an image of a pharmacy in France, when the model has only seen pharmacies  in the US (both will look very different, there cannot be a reasonable expectation for such a transfer). On the other hand, data augmentation can strongly abstract away certain image features, it thus is reasonable that it improves performance for ImageNet-R, but is limited for the other two datasets.

The core issue seems to lie in the construction of the datasets:

1) We know that deep networks have a texture bias, so there is no reasonable expectation for transfer to ImageNet-R. There wouldn't be any expectations to improve this performance for any change, but enlargement of the dataset towards more abstract depictions of the objects. This seems to be confirmed by Table 1, where abstract augmentations (e.g. enlarging the training set with samples that are  in some respect more similar to the test set) clearly improves results, but simple architectural changes or simple augmentations do not help. This would support the hypothesis that to improve robustness, more similar data is necessary (i.e. the training set simply doesn't sufficiently cover the space of images that we expect the model to perform on).

2) The SVSF dataset paints a similar picture. Small shifts (e.g. images taken a year apart) hardly influence performance, whereas a extreme shift (e.g. in location) breaks the model even with the augmentations. This seems reasonable, as the augmentations certainly don't cover the shift that typically happen for building appearances between continents.

3) DFR paints a similar picture: Abstract augmentation slightly helps if it roughly matches the shift. Other simple augmentations or architectural changes do not significantly change results.

My conclusion from the experiments would be something that is well known:  enlarge the dataset to better cover what you expect your model to do. If this can be done with automatic augmentations  (e.g. for zoom, some augmentations that are closer to artistic renditions) then you can use these augmentations. If not: collect more data. It is thus not clear what the provided analysis provides on top of this.

--- Other ---

Some clarifying questions:

- Why is SVSF limited to augmentations? What does a 30 day retention window mean?
- Can you clarify you conclusions on DFR? Why don't you see evidence for larger models or pretraining? Both seem to substantially improve performance?

--- Summary ---

I think this work is interesting and is in principle asking the right questions. However, the analysis and conclusions currently do not providing robust and generalizable insights that advance the field.

--- Post rebuttal ---

I'm keeping my initial score. My concern remains (and is apparently reflected by R1): the datasets and results do not allow to draw clear conclusions. The paper overall furthers our understanding on robustness only in a very limited way. The new dataset adds another specialized dataset to the mix. I disagree that the community should first exhaustively and randomly add datasets to the literature without coming up with a definition of robustness or at least try to categorize. The authors in their rebuttal criticize the community that they are looking at robustness and distribution shifts in a too simplistic way, but at the same time the presented work doesn't make an effort to change this either.

To close the remaining question:

- "Could you elaborate? We know that humans and primates can generalize to new renditions that they have not seen before (Itakura, 1994; Tanaka, 2006), while some other species cannot. Consequently more than training data matters."

With "extreme", I mean distributions shifts that keep semantics, but change appearance strongly. If we go as far looking at biological systems, then yes, it is not only about training data. There is an additional mechanism at play that we don't know and currently can't replicate in ML. Given our current understanding, it is presumptuous to suggest that NN architectures and training approaches as they are covered in this work will be able to do these kinds of generalizations at the level of humans or primates.

- "The empirical reality does not currently allow for a simple, single-cause characterization of robustness"

I completely agree to this statement. However, this doesn't mean that characterization of robustness cannot be done by taking into account multiple factors systematically.

I acknowledge that collecting a new dataset is a non-trivial effort and can be useful.  I acknowledge that the paper proposes an additional augmentation technique that seems to improve results in certain cases. All factors together taken together lead to my final score.

---

> ### Author Response · Authors · 2020-11-17
> **Larger Models and Self-Attention Can Sometimes Help Too**
>
> Thank you for your careful analysis of our paper.
>
> + “Some hypotheses are supported by some datasets, others are not. No clear conclusions that hold across datasets can be drawn.”
>
> A main point of the paper is that there are currently no silver bullets for robustness, and no previously proposed method consistently works. Many papers are claiming to “improve robustness” and treat improving performance on one benchmark as sufficient evidence. This shows scientific standards in this research area must change, and we argue that they must target multiple metrics not just one scalar.
>
> + “Why should, for example, self-attention help to improve performance on artistic renditions, something that is very different from a blurred variant of an image?”
>
> As it happens, self-attention greatly improved performance on 75 unseen corruptions and even real-world “natural adversarial examples.” This is why self-attention was previously thought to “improve robustness.” It was not at all obvious that self-attention would consistently reduce robustness on renditions. Our work is showing that the picture is much more complicated and less predictable, so evaluation should widen to mitigate false positive findings in future research.
>
> + “but simple architectural changes or simple augmentations do not help.”
>
> In Figure 6, we show that architectural changes such as making the network larger do help on ImageNet-R. For reasons like this, we think that the results are not completely predictable.
>
> + “Small shifts (e.g. images taken a year apart) hardly influence performance”
>
> In DeepFasion Remixed we can observe several seemingly small geometric shifts that greatly change performance.
>
> + “Why is SVSF limited to augmentations? What does a 30 day retention window mean?”
>
> SVSF 30 day lifespan in order to comply with data policy.
> This is because SVSF updates monthly (which is the 30 day retention window), so we were only able to fully train five large-scale SVSF classification models within that time limited span. In contrast, ImageNet-R, DeepFashion Remix, and Real Blurry Images are fixed datasets.
>
> + “Why don't you see evidence for larger models or pretraining?”
>
> When constructing summary table 4, we were considering the IID/OOD gap so that we could identify methods which improve robustness beyond just increasing IID accuracy.
>
> + “If this can be done with automatic augmentations (e.g. for zoom, some augmentations that are closer to artistic renditions) then you can use these augmentations.”
>
> How best to do this is not obvious, and often shifts are not known beforehand. For instance, DeepAugment does similarly or better on the real-world occlusion shift in DeepFashion Remixed than Random Erasure, which directly simulates occlusion.
>
> + “My conclusion from the experiments would be something that is well known: enlarge the dataset to better cover what you expect your model to do.”
>
> Perhaps we could formulate “My conclusion from the experiments would be something that is well known: enlarge the dataset to better cover what you expect your model to do” as a hypothesis, maybe the “train-test discrepancy is all that matters" hypothesis.
> We agree that reducing distribution shift is one way to improve robustness, but on ImageNet-C, ImageNet-A, and ImageNet-R, _Larger Models_ helped 2/3 times, _Self-Attention_ helped 2/3 times, and _Diverse Data Augmentation_ helped 2/3 times. Hence factors beyond reducing the train-test discrepancy can help robustness. Hopefully our analysis helped weigh in on this hypothesis.
>
> We do not think our results are obvious conventional wisdom. Before we submitted the paper, for fun we ran a small informal forecasting competition where we asked our colleagues to predict model accuracy on these new datasets. Results varied wildly. For example, the first three authors did not expect AugMix to help with ImageNet-R, while the senior authors did. Some respondents expected pretraining to nearly solve ImageNet-R, but of course ImageNet-21K pretraining reduced the IID/OOD gap by only 0.2%. Some expected SVSF’s temporal shift to greatly degrade accuracy, possibly given previous claims about the extreme sensitivity of image classifiers. Although hindsight is 20/20, the results are not easy to predict before the fact. We didn’t think to include the survey results since it was informal and included only around 10 researchers, but we do not think all results were predictable given just the train-test discrepancy.
> Finally, our paper includes more than our results: (1) we introduce DeepAugment which gets sets a new state-of-the-art for ImageNet-C and ImageNet-R, (2) we assemble evidence that robustness research practices are far too narrow and must change, and (3) we collect many novel datasets to advance future robustness research.
>
> We hope we were able to address your valid questions and we thank you for your helpful suggestions. Do you have any remaining concerns?

---

> > ### Comment · AnonReviewer2 · 2020-11-24
> > **Thanks for the response**
> >
> > Thanks for the response and revisions. Apologies for my late reply.
> >
> > While the response addresses some of my initial questions, I still think that my main point stands: the conclusions that can be drawn are not clear.  I do see the value in adding a dataset, and I also see value in highlighting that current practices might not be sufficient.  However, there is a large difference between simple transformations (like blur, noise, etc.) where it seems possible that architectural modifications, attention, etc. make a significant difference for robustness. However, for extreme (even semantic) modifications, like the ones shown in ImageNet-R, it is unlikely that there is a simple architectural modification that can close this significant gap.  After all, much of this is context-dependent. Is a rendition of an object really the object, or something entirely new? Of course it depends on the circumstances and what we are asking the model to do. So it seems impossible to answer this questions, without answering this question for the model during training. Taking different types of disturbances together, it is of course possible to make the statement that certain modifications, like self-attention, help in 2/3 cases, but it doesn't give a clear picture of what is going on.
> >
> > As for the restated “train-test discrepancy is all that matters" hypothesis: It be inclined to say that this is true for the "extreme" form of robustness. This seems to be also shown by Table 7 in the supplement, where WSL-pretraining has he biggest influence by far. WSL is likely to contain some form of renditions since it based on web-scraping and hashtags.
> >
> > I see that the "rigorous" was dropped from the title. A rigorous evaluation of robustness would would need a better definition of robustness. Some taxonomy that clearly distinguishes robustness to minor pixel-level transformation from true semantic robustness, and everything in between. This would also allow combinations and thus cover the stated multivariate nature. A clear distinction between types of disturbances, would also allow to make clear statements about robustness, which need to be conditioned on the type of distortions that one is likely to encounter.

---

> > > ### Author Response · Authors · 2020-11-24
> > > **Nuance is Key**
> > >
> > > Thank you for your response and for reviewing our changes.
> > >
> > > + "However, there is a large difference between simple transformations (like blur, noise, etc.) where it seems possible that architectural modifications, attention, etc. make a significant difference for robustness. However, for extreme (even semantic) modifications, like the ones shown in ImageNet-R, it is unlikely that there is a simple architectural modification that can close this significant gap."
> > >
> > > Recall that ImageNet-A has images that are completely natural and not simple transformations, and they represent a difficult distribution shift. We note that larger models do help with both ImageNet-R and ImageNet-A. Also, self-attention helps with ImageNet-A. We're observing that architecture can play a role.
> > > Moreover, on ImageNet-A, a ResNet-152 gets 6.05% accuracy; changing the ResBlocks with Res2Net blocks (an architectural change) results in 22.40% accuracy--the accuracy almost quadrupled due an architectural change.
> > > Consequently, architectural changes which were not designed for robustness can turn out to help with robustness. If the community were to search for architectures that are more robust, then gains could be even larger. Evidently simple architectural changes can sometimes greatly increase robustness.
> > >
> > > + "As for the restated “train-test discrepancy is all that matters" hypothesis: It be inclined to say that this is true for the "extreme" form of robustness."
> > >
> > > Could you elaborate? We know that humans and primates can generalize to new renditions that they have not seen before (Itakura, 1994; Tanaka, 2006), while some other species cannot. Consequently more than training data matters.
> > >
> > > + "Some taxonomy that clearly distinguishes robustness to minor pixel-level transformation from true semantic robustness, and everything in between."
> > >
> > > We agree the community should work toward a taxonomy. To avoid prematurely systematizing distribution shifts with a precise taxonomy, the community should first uncover and analyze new types of distribution shifts. In our paper, we begin a concerted effort toward this end by analyzing numerous new distribution shifts.
> > >
> > > + "but it doesn't give a clear picture of what is going on."
> > >
> > > The empirical reality does not currently allow for a simple, single-cause characterization of robustness. Previous works often treat robustness as though it was is this simple, but we argue more nuance is needed. That said, our results are not overly pessimistic. Our work is has new explanatory power: we now observe that the textural hypothesis has more evidence than previously thought, while other hypotheses are less tenable. Likewise, we identify that some methods are more useful than previously thought. For instance, it was argued that data augmentation could not help with any real-world distribution shifts, which our Real Blurry Images and ImageNet-R datasets debunk. While we show that robustness less simple than previously thought, the community ought to know.
> > >
> > > We hope we have clarified and addressed the thrust of your concerns. Do you have any remaining questions?

---

### Official Review · AnonReviewer1 · 2020-10-28
**The unclear motivation for three different benchmarks and DeepAugment**

**Rating:** 4
**Confidence:** 5

**Review:**

This paper proposes three new benchmarks for robustness, named ImageNet-R, StreetView StoreFronts, and DeepFashion Remixed. Also, this paper proposes a new augmentation called DeepAugment.

**Pros**

\+ A new benchmark could be useful for many researchers in this field.

**Cons**

**[How this paper solve the seven motivations is unclear]**
To me, the seven motivations (larger model increases the robustness, self-attention increases the robustness, diverse augmentation increases the robustness, pretrain models increase the robustness, texture bias harms the robustness, IID dataset determines the robustness, synthetic robustness is not helpful for the real-world robustness) *are completely independent to each other*. After I read the paper, it is still remaining as a question of how the seven motivations are related and how they are solved by this paper.

First of all, what does "robustness" mean in this paper? For instance, adversarial robustness represents the error rate against the worst-case attacks in the L-p ball of the given input. However, in this paper, I feel the terminology "robustness" is ill-defined.

Second, if this paper argues that "larger model" or "self-attention" increases the "robustness", I would expect

- the rigorous definition of the robustness
- the theoretical guarantee or strong empirical evidence that a larger model or self-attention can increase the robustness against the proposed threat model.

However, I cannot find any detail in this paper.

Also, texture robustness or synthetic robustness is not fully discussed in this paper. Why we have to consider them? And how the proposed benchmarks support the arguments?


**[Motivation to a new benchmark is not enough]**
There are already many ImageNet benchmarks including ImageNet-C, ImageNet-P [1], ImageNet-A, ImageNet-O [2], ImageNet-V2 [3], clean label [4], stylized ImageNet [5] and other possible benchmarks.

[1] Hendrycks, Dan, and Thomas Dietterich. "Benchmarking neural network robustness to common corruptions and perturbations." ICLR 2019
[2] Hendrycks, Dan, et al. "Natural adversarial examples." arXiv preprint arXiv:1907.07174 (2019).
[3] Recht, Benjamin, et al. "Do imagenet classifiers generalize to imagenet?." ICML 2020
[4] Beyer, Lucas, et al. "Are we done with ImageNet?." arXiv preprint arXiv:2006.07159 (2020).
[5] Geirhos, Robert, et al. "ImageNet-trained CNNs are biased towards texture; increasing shape bias improves accuracy and robustness." ICLR 2019

I personally cannot find any motivation to use ImageNet-R instead of the above benchmarks to evaluate the robustness.
Especially, I feel the first seven motivations are independent to the ImageNet-R.

Even if we consider ImageNet-R as a new ImageNet "robustness" benchmark,
I still wonder why we have to evaluate our models to ImageNet-R instead of ImageNet-C, -P, -A, -V2, and clean labels.

Furthermore, I would like to cite a recent paper on measuring robustness to distribution shift in ImageNet classification (a contemporary work)
[6] Taori, Rohan, et al. "Measuring robustness to natural distribution shifts in image classification." arXiv preprint arXiv:2007.00644 (2020).
This paper shows that natural robustness is completely relying on the true test set accuracy.
If we employ a new robustness benchmark, I would expect a rigorous reason why we need a new benchmark.

**[ImageNet-R: Ambiguity on selecting 200 classes and the rendition classes]**
The most ambiguous thing in this paper is "200 sub-classes" to collect the dataset (as section 3.1). Why the authors use 200 sub-classes instead of the original 1,000 classes?
How the authors choose "200 sub-classes" from the 1000 classes?

Furthermore, this paper aims to solve the robustness problem (where the ``robustness'' is not defined in this paper). I wonder how the ImageNet-R benchmark can evaluate the robustness of the trained models.

Also, I wonder how the "renditions" are chosen. We always can collect a new ImageNet benchmark by crawling a new dataset from the web.
If the renditions in ImageNet-R cannot ensure the whole "renditions" in the real-world, we also can suffer from the overfitting issue to the proposed ImageNet-R.
It could be problematic because a deep model is known to not be able to generalize to the unseen noises [7]

[7] Geirhos, Robert, et al. "Generalisation in humans and deep neural networks." Advances in neural information processing systems. 2018.

**[Why SVSF and DeepFashion Remixed datasets are required to support the original seven motivations?]**
Although I like to test a new benchmark for testing different methods, it is not clear why SVSF and DeepFashion remixed datasets are required to support the original motivations. Why ImageNet-R is not enough? How ImageNet-R, SVSF, and DeepFashion revisited are related?
I feel the proposed three benchmarks are not really related.

**[Can not find any criteria to build the proposed dataset]**
I believe this paper aims to solve a "robustness" problem. However, I cannot find any scientific protocol which supports that "achieving high accuracy on the proposed benchmarks truly solves the robustness problem". How the datasets are built? How we can trust the benchmark, while other possible benchmarks are not reliable?

**[DeepAugment details are missed in the main paper]**
In my opinion, this paper has two contributions (1) a new benchmark, and (2) a new augmentation method to solve (1). However, the augmentation method (DeepAugment) details are not able to understand without reading the appendix.
Even after I read the appendix, I still cannot understand the motivation of the DeepAugment and the method details.
Why do we need to apply layer-wise distortions instead of input-level distortions? How the distortions are chosen? Are the distortions independently chosen from the benchmark distortions? I still have many questions.
Furthermore, I feel that DeepAugment requires a lot of hyperparameters, especially for choosing the "distortions". I wonder how the authors choose the hyperparameters, especially the set of "distortions". If the authors directly tune their method on the ImageNet-R, it is not fair and not convincing benchmark to evaluate the robustness.

---

**Final review**

After reading the paper, other reviews, and author responses carefully again, I decided to remain on the rejection side because

- I think the proposed dataset does not really guarantee the robustness against "real-world distribution shifts" because
  - This paper did not rigorously define what the "real-world distribution shifts" are. In the final response, the authors mentioned that *"It is clear that temporal, hardware, geographic, and rendition shifts occur in the real world."*, but to me, it is not clear whether they are really common and representative in the real-world deployment scenario and really threaten deep models.
  - Because the real-world distribution shift is not well-defined here, I feel the "robustness" is also ill-defined too. According to the author response, robustness is defined as the accuracy gap between "in-distributed" samples and "out-of-distributed" samples (not critical, but OOD is usually defined as the same data distribution, but unseen class. I think this terminology need to be polished). However, here OOD (distribution shift) is ill-defined, and the robustness test is heavily dependent on the test dataset.
  - Thus, if we just test the "robustness against real-world distribution shift" on the proposed dataset only, it can lead to wrong conclusions, e.g., assume we have a model can be specifically better in a specific shift, e.g., rendition shift, but not generalized to other shifts, then ImageNet-R benchmark cannot measure how this model is vulnerable to the other shifts. It will confuse researchers in this field. Hence, I think this paper needs more justification for the new dataset (e.g., why the chosen shifts? why 200 classes for ImageNet-R? why different three datasets?), and need more human studies (e.g., humans can correctly classify the shifted images and non-shifted images) such as [3, 5, 7, 8].
- This paper is not clearly presented. After reading the paper, I am still confusing about how to understand the experimental results. To me, the benchmark results cannot answer these questions well. I think R2 has a similar opinion on me in this criterion.
- It is not mentioned in my previous reviews, so I lower the weights for this part to the final decision, but there are already some datasets benchmarking the dataset distribution shifts, e.g., PACS [9], NICO [10]. It may not be true that this kind of distribution shift is only measurable by the proposed datasets. But, as my first words, I noticed that I did not mention these datasets in my previous reviews, and these datasets will not affect my review a lot.
  - https://domaingeneralization.github.io/
  - http://nico.thumedialab.com/

[8] Shankar, Vaishaal, et al. "Evaluating machine accuracy on imagenet." International Conference on Machine Learning. PMLR, 2020.
[9] Li, Da, et al. "Deeper, broader and artier domain generalization." Proceedings of the IEEE international conference on computer vision. 2017.
[10] He, Yue, Zheyan Shen, and Peng Cui. "Towards Non-IID Image Classification: A Dataset and Baselines." Pattern Recognition (2020): 107383.

Of course, building a new dataset is a non-trivial effort, and measuring real-world robustness is not an easy task (maybe it even can be an impossible task). However, I think this paper can not clearly present how the proposed benchmark can solve the real-world distribution shifts and how can we move forward in the next directions.

To sum up, I think this paper is okay, but not enough to be accepted to ICLR main conference paper. However, I will respect all decisions made by AC.

---

> ### Author Response · Authors · 2020-11-17
> **Reply (2/2)**
>
> + “Why do the authors use 200 sub-classes instead of the original 1,000 classes?”
>
> Determining the true class of a rendition image among all 1000 classes can be difficult. Annotators would have to accurately distinguish between sculptures of Norwich terriers and Norfolk terriers. In addition to selecting sub-classes which could be reliably annotated, we also chose classes with enough pictures online to form a representative sample. For instance, it is much easier to find paintings of strawberries than paintings of radiators. We settled on the number 200 following previous work such as ImageNet-A.
>
> + “We can always collect a new ImageNet benchmark by crawling a new dataset from the web.”
>
> We believe collecting new datasets holds academic value and should not be downplayed. Annotating a dataset is nontrivial and costs money and time.
>
> + “If the renditions in ImageNet-R cannot ensure the whole "renditions" in the real-world, we also can suffer from the overfitting issue”
>
> We claim that the addition of Imagenet-R allows for benchmarking on new forms of shift that were previously not possible (Stylized ImageNet includes one synthetic rendition type, while we test dozens of real rendition types). Our study shows that some forms of generalization do occur. This is clear from our DeepAugment results which generalizes to many unseen forms of shifts such as ImageNet-C, ImageNet-R, and the Real Blurry Images.
>
> + “It is not clear why SVSF and DeepFashion remixed datasets are required to support the original motivations. Why is ImageNet-R not enough?”
>
> We collected SVSF and DeepFashion Remixed in order to thoroughly test the various hypotheses presented in the literature. These two datasets contain new distribution shifts not present within ImageNet-R or prior datasets and also provided new evidence against several hypotheses which were not apparent from results on ImageNet-R alone. Indeed we found no prior method which improves robustness on SVSF and DeepFashion, which is in contrast to Imagenet-R.
>
> + “I cannot find any scientific protocol which supports that ‘achieving high accuracy on the proposed benchmarks truly solves the robustness problem’”
>
> We did not make any claims of solving robustness in our paper. However, we would claim that higher accuracy on the unseen distribution shifts in this paper constitutes empirical evidence of increased robustness to a broader class of shifts. We found evidence that previous benchmarks like ImageNet-C are indicative of nontrivial generalization to some forms of unseen shifts, but analysis solely based on this one dataset cannot draw broad conclusions about robustness.
>
> + “Are the distortions independently chosen from the benchmark distortions?”
>
> Thank you for raising this point. We have added another paragraph to the DeepAugment section in the main body to clarify. None of our distortions overlap with real-world ImageNet-R renditions, and we do not tune on any ImageNet-R images (otherwise it would not be a distribution shift). We provided pseudocode in the paper, and we provide complete code in the supplementary materials.
>
> + “If the authors directly tune their method on the ImageNet-R, it is not fair and not convincing benchmark to evaluate the robustness.”
>
> In the revised paper, we demonstrate that high ImageNet-R performance is possible using random (untuned, unlearned) architectures, so our technique is flexible and does not require careful tuning. DeepAugment also works with randomly sampled neural architecture search architectures. https://openreview.net/pdf?id=o20_NVA92tK#page=20 This demonstrates the versatility of DeepAugment, as we can generate useful deep augmentations from randomly sampled architectures with random weights. Thank you for suggesting adding this type of sensitivity analysis.
>
> We ask that you reconsider your assessment of our paper as we think we have defended that there is more than one pro to our paper, such as the pros listed by other reviewers (DeepAugment, comprehensive experiments, our Real Blurry Images result and its implication for ImageNet-C, our various new datasets, etc.).
>
> We hope we were able to address your valid questions and we thank you for your helpful suggestions. Do you have any remaining concerns?

---

> > ### Comment · AnonReviewer1 · 2020-11-24
> > **Thanks for many clarifications**
> >
> > Thanks for the detailed answers to my questions.
> >
> > First, please clarify the details in the answers to the revised paper later (I am okay to revise the paper after the review period, I am sorry for my late reply)
> >
> > Second, I want to say thanks to the authors for putting a lot of effort to build new datasets. I absolutely agree that "annotating a dataset is nontrivial and costs money and time", and really appreciate building the datasets. However, even though dataset collection requires non-trivial effort, it does not mean all new datasets hold academic value.
> >
> > Especially, I still have trouble to find a solid conclusion for this paper. As far as I understand correctly, this paper proposes **"rigorous"** evaluation of real-world distribution shifts. The answers to my questions help my understanding a little bit, but I still have trouble to find scientific reasons and protocols for how the proposed datasets provide a **rigorous evaluation** of **real-world distribution shifts**.
> >
> > Furthermore, I am hesitating to say the proposed datasets can really represent "real-world distribution shifts". I totally agree with the rebuttal:
> >
> > > the addition of Imagenet-R allows for benchmarking on new forms of shift that were previously not possible
> >
> > but it does not guarantee ImageNet-R really covers all possible real-world distribution shifts, or at least the proposed dataset covers "quite a lot" real-world distribution shifts. In my opinion, it is due to the unclear definition of "distribution shift" in this paper. Without a solid definition of the real-world distribution shift, I think that similar questions will appear repeatedly, and a new "ImageNet-R" will be proposed. It will confuse researchers in this field.
> >
> > Also, it is still unclear how 200 sub-classes are sampled from the ImageNet 1000 classes. I understand collecting 1000 different images is costly and difficult, but I think we need some solid protocol to choose the proposed 200 sub-classes, e.g., the 200 sub-classes can cover 1000 classes in the wordnet structure.
> >
> > Thanks for many clarifications, and thanks for your datasets, but I still incline to the rejection side.

---

> > > ### Author Response · Authors · 2020-11-24
> > > **Paper Renamed**
> > >
> > > Thank you for your response and for reviewing our changes.
> > >
> > > + "please clarify the details in the answers to the revised paper later"
> > >
> > > We have added many of these clarifications in the revised paper. All of these changes can be found through "Show Revisions" > "Compare Revisions." For instance, our clarification about 200 classes are in the paper.
> > > "We choose a subset of the ImageNet-1K classes, following Hendrycks et al. (2019b), for several reasons. A handful ImageNet classes already have many renditions, such as “triceratops.” We also choose a subset so that model misclassifications are egregious and to reduce label noise. The 200 class subset was also chosen based on rendition prevalence, as “strawberry” renditions were easier to obtain than “radiator” renditions. Were we to use all 1,000 ImageNet classes, annotators would be pressed to distinguish between Norwich terrier renditions as Norfolk terrier renditions, which is difficult." As evident in Appendix B, our 200 classes cover the main superconcepts in WordNet that ImageNet-1K covers. Additionally, our DeepAugment architecture sensitivity experiments, spanning pages 20-22, were added in the revision.
> > >
> > > + "but I still have trouble to find scientific reasons and protocols for how the proposed datasets provide a rigorous evaluation of real-world distribution shifts"
> > >
> > > It is clear that temporal, hardware, geographic, and rendition shifts occur in the real world. Hence the contention might be about the word "rigorous"; we did not intend to suggest our paper is "theoretically rigorous." We have changed the paper title to "A Critical Analysis of Distribution Shift" due to your concern.
> > >
> > > Our paper's conclusion is that future work must abandon current practices. Our paper shows evaluation is far too narrow and methods often do not work in new regimes, so to prevent false positive findings, future work needs to evaluate more broadly than before. We are calling for new evaluation standards and our experiments demonstrate their necessity.
> > >
> > > + "but it does not guarantee ImageNet-R really covers all possible real-world distribution shifts"
> > >
> > > We cover far more distribution shifts than previous work (Style Transfer vs. art, cartoons, deviantart, graffiti, embroidery, graphics, origami, paintings, patterns, plastic objects, plush objects, sculptures, sketches, tattoos, toys, video game renditions, ...).
> > > We do not think any image classification benchmark could ever cover all possible distribution shifts since that is an unattainable standard for any real dataset. If a network can generalize to numerous unseen renditions, that is evidence that it is robust to rendition shifts. However, we agree that it does not provide absolute proof, but no other real distribution shift datasets have this property.
> > >
> > > Thank you for your engagement during the discussion period. Do you have any other questions?

---

> ### Author Response · Authors · 2020-11-17
> **Reply (1/2)**
>
> Thank you for your careful analysis of our paper. We have attempted to address your questions below.
>
> + “How are the seven motivations related?”
>
> The seven hypotheses that we consider are previously proposed _methods_ for improving robustness or previously proposed _properties_ about robustness. Some hypotheses are especially related, such as Texture Bias and Diverse Data Augmentation since the latter attempts can help ameliorate the former. There are seven hypotheses analyzed because we wanted our analysis to stress test previous proposals comprehensively.
>
> + “What does ‘robustness’ mean?”
>
> We define robustness as held out test set accuracy, where the test set is understood to come from a distribution that can be in many ways different from the training distribution. Symbolically, if we assume $f$ is our classifier, then robustness is
> $1 - E_{(x,y)\sim D_{\text{test}}} [l_{0/1}(f(x), y)].$
> To control for in-distribution accuracy, we also look at the IID/OOD gap throughout the paper, which is
> $E_{(x,y)\sim D_{\text{test}}} [l_{0/1}(f(x), y)] - E_{(x,y)\sim D_{\text{IID test}}} [l_{0/1}(f(x), y)].$
> For the ImageNet-C task, we compute the mCE which has a very similar formula.
>
> Note that the space of distributions which differ from the training distribution is multifaceted, and thus there is no single type of robustness or single existing benchmark which is representative of all forms of robustness. We cannot expect methods or hypotheses targeting one form of robustness (or one class of distributions) to generalize to all potential shifts. This necessitates the development of increasingly diverse sets of robustness benchmarks so that we can begin to understand and categorize the various forms of shift that can occur at test time.
>
> + “texture robustness or synthetic robustness is not fully discussed in this paper. Why we have to consider them?”
>
> We assess robustness to synthetic distortions with the ImageNet-C dataset. The motivation for assessing performance on ImageNet-C arises because models which can generalize to various unseen distortions are desirable in several application areas such as autonomous driving. Likewise, ImageNet-R enables us to estimate texture robustness, so we discuss synthetic and textural robustness in the paper. Humans can generalize to unseen many distortions, but current models are fragile, so current models are demonstrably subhuman.
>
> + “There are already many ImageNet benchmarks including ImageNet-C, ImageNet-P [1], ImageNet-A, ImageNet-O [2], ImageNet-V2 [3], clean label [4], stylized ImageNet [5]”
>
> We analyze several of these datasets. We should like to note that we focus on classification accuracy in this paper, so we do not look at classification stability as in ImageNet-P. ImageNet-O is for testing uncertainty estimation not classification. We did not consider ImageNet-V2 due to Engstrom et al., 2020, as we indicate in the related work. Clean/ReaL ImageNet labels are not for testing robustness to input distribution shift since the input images are identical to the original ImageNet images. Stylized ImageNet is synthetic and tests one rendition type, while we test over a dozen real-world renditions styles with ImageNet-R.
>
> The SVSF dataset has distribution shifts not yet systematically measured in the literature: hardware shift, temporal shift, geographic shift. These distribution shifts are of clear practical importance.
>
> The Real Blurry Images dataset is the first ImageNet-like dataset with real blurs and provides critical information about the external validity of ImageNet-C.
> The DeepFashion dataset has highly controlled distribution shifts (occlusion, size, viewpoint, zoom) not covered in your list above.
>
> We collected these novel datasets to see how well previous hypotheses in the literature hold up under new conditions. We found that numerous datasets are necessary for studying robustness. While many previous works focus on one dataset, we collected many datasets and found they yield highly varied results. Hence our new datasets enable researchers to test their hypotheses across many new distribution shifts.
>
> Our datasets (1) test new distribution shifts, (2) enabled new conclusions about robustness, and (3) are sometimes harder than previous datasets. Hopefully this provides some motivation.
>
> + “I personally cannot find any motivation to use ImageNet-R instead of the above benchmarks to evaluate the robustness.”
>
> The closest analogue is Stylized ImageNet, which uses style transfer to generate images. This specific rendition is synthetic and tests one rendition style/one corruption type. With ImageNet-R, we can test over a dozen real-world renditions styles. Hence ImageNet-R tests multiple real-world distribution shifts, so it is more rigorous and has more _external validity_ compared to synthetic style transfer images.

---

> ### Comment · Area_Chair1 · 2020-11-22
> **request for response**
>
> Greetings R1,
>
> You have received detailed responses to your question. Would you mind taking a look and letting us know what you think.
>
> Thanks

---

### Official Review · AnonReviewer3 · 2020-10-29
**An empirical evaluation of natural distributional shifts in image classification**

**Rating:** 7
**Confidence:** 5

**Review:**

This paper contributes three new datasets to evaluating seven robustness hypotheses.

Strengths:

1.The introduced three databases would be valuable to probe the generalization of classifiers in the real world.

2. The deep augmentation method seems neat. It is a unified method to produce a variety of perturbations (despite in a less controllable way). And the performance gap on the proposed databases is noticeably reduced by DeepAugment.

Weaknesses:

1. The authors may clearly state how they collect human labels for these datasets. In practice, collecting 1 out of 200 possible labels in ImageNet-R is not trivial.

Other comments:
1. What is the intended solution for solving the StreetView StoreFronts dataset? Is it the structure of the store or just text in the image?
2. As shown in Fig. 3, DeepAugment seems to distort the input images. Is there a way to systematically control the distortion levels? Does the architecture of the autoencoder matter?
3. There is another line of research to construct small but adaptive datasets to probe the generalizability of classifiers [C1], published in ICLR2020. The authors may want to be aware of it.

[R1] I Am Going MAD: Maximum Discrepancy Competition for Comparing Classifiers Adaptively, https://openreview.net/forum?id=rJehNT4YPr

---

> ### Author Response · Authors · 2020-11-17
> **ImageNet-R Clarification**
>
> Thank you for your careful analysis of our paper and your positive review!
>
> In our revision, the length of our ImageNet-R data collection description has doubled. We queried Flickr with strings including “lighthouse cartoon.” Then MTurk annotators selected true positive lighthouse renditions from all the images associated with the query “lighthouse cartoon.” As a second quality control, graduate students manually filter the resulting images and ensure that individual images have correct labels and do not contain multiple labels.
>
> “Is there a way to systematically control the distortion levels? Does the architecture of the autoencoder matter?” DeepAugment produces very diverse outputs so minor changes can produce very different augmentations, but the images retain semantic content and can improve learned representations. We were able to use two different architectures (EDSR and CAE), so the technique can work with different architectures.
>
> We also found it to work with randomly constructed architectures with Res2Net blocks as well (like what is found in the NAS literature), so the technique is flexible, not customized to a specific architecture, and not ad-hoc. In the final figure of the revised paper, we show an example of control augmentation strength with a single parameter. This is in the revised paper: https://openreview.net/pdf?id=o20_NVA92tK#page=20
>
> Thank you for bringing “I Am Going MAD: Maximum Discrepancy Competition for Comparing Classifiers Adaptively” by Haotao Wang, Tianlong Chen, Zhangyang Wang, Kede Ma, ICLR 2020 to our attention. We focus on fixed datasets since that is the existing standard, but adaptive datasets is a promising direction. We have cited it and will look at it more closely in future experiments.

---

> > ### Comment · Area_Chair1 · 2020-11-22
> > **comment on data collection**
> >
> > Greetings R3,
> >
> > The authors have updated the text to address your question for more information about data collection. Can you please take a look and let us know if it addresses your question. Please also take a look at the other reviews.
> >
> > Thanks

---

> > ### Comment · AnonReviewer3 · 2020-11-25
> > **Regarding Motivation**
> >
> > Thank the authors for taking my comments into consideration. I've also taken a closer look at other reviewers' comments, and agree with Reviewer #1 that the establishment of the dataset is less motivated, and does not distinguish itself much from many other post-ImageNet datasets for robustness testing. Moreover, the DeepAugment technique seems a bit ad hoc and less controllable. However, according to the experimental results, the proposed datasets have already shown their academic values, down-weighting some techniques for robustness purposes, and exposing the failures of existing methods. Therefore, I will stick to my current score.
> >
> > Nevertheless, I have a major concern about the trend of building $\textbf{pre-fixed}$ and $\textbf{pre-selected}$ datasets (including the proposed ones) in image classification and other subfields of computer vision. The procedure is time-consuming and less efficient. Taking ImageNet-A as an example.  It was published in 2019, and algorithms in 2020 already achieve excellent performance on it, making this dataset built with many human and financial resources less useful. Similar arguments may apply to the proposed three datasets. To the reviewer's biased view, it is important to start to think when building a new dataset, how to select the images to label. One plausible way is to select $\textbf{diverse}$ images that have the greatest potentials to $\textbf{rank}$ the competing methods so as to minimize the human labeling effort. That is, we may need a hybrid measure of (algorithm-dependent) diversity and rank efficiency to guide the image selection process. Otherwise, like the proposed datasets, even we create many hard images to falsify existing methods, they may correspond to the same underlying root causes and are less diverse, wasting the human labeling budget.

---

### Decision · Program_Chairs · 2021-01-07
**Final Decision**

**Decision:**

Reject

**Comment:**

The authors propose a new dataset to evaluate the robustness of image classifiers. The dataset consists of data from three sources: a crowdsourced dataset collected by the authors called ImageNet-Renditions, images from Google street view, and data sampled from DeepFashion2. This new dataset allows the authors to test robustness to different renditions of an object (e.g. artistic depictions of an object category) and robustness to changes in geography and camera type. In addition, they propose a new augmentation strategy called DeepAugment which consists of an encoder/decoder style network that transforms the appearance of the input image by simply applying different random perturbations of the weights of the augment network. Robustness results are presented on the previously described datasets where the proposed augmentation strategy in combination with an existing approach (AugMix) performs best in some cases. However, the results are not convincing and AugMix often outperforms the new method.

In general, the authors did a good job addressing many of the comments (e.g. they provided more detail about how ImageNet-R was collected), but there were still several lingering concerns. R4 was the most positive about the paper, but unfortunately was one of the least vocal during the discussion. R1 was concerned that the paper did not do a great job of defining what was meant by robustness. This AC doesn't agree fully with their concerns, but does agree that more care could have been taken to position the paper better in light of the existing datasets that are already available (see R1’s comments). As the reviewers and authors note, collecting new datasets is a lot of work so care should be taken to ensure that this is not duplicate effort. The authors addressed these concerns in their response to some extent, but more discussion is needed in the paper.

There was a lot of discussion between the authors and reviewers about this paper. The new dataset has a lot of merit, but there is some concern that the paper does not do a great job of clearly presenting its findings and conclusions. In addition, the proposed augmentation technique is slightly underwhelming performance wise and not very clearly described in the main paper. R2 sums up the opinion of this AC: “I think this work is interesting and is in principle asking the right questions. However, the analysis and conclusions currently do not provide robust and generalizable insights that advance the field.” There is clearly a lot of promise here, and the current recommendation is a weak reject. The authors are strongly encouraged to take the detailed feedback they have received on board and to revise the paper to further improve it for a future submission.